# Food and Monastic Space: From Routine Dining to Sacred Worship—Comparative Review of Han Buddhist and Cistercian Monasteries Using Guoqing Si and Poblet Monastery as Detailed Case Studies

**Weiqiao Wang**

College of Architecture and Urban Planning, Tongji University, Shanghai 200092, China;
wangweiqiao@tongji.edu.cn

**Abstract:** Through an exploration of meal regulations, dining rituals, and monastic rules of Han Buddhist and Cistercian monks, this article discusses how food affects space formation, layout organization, and site selection in monastic venues using Guoqing Si and Poblet Monastery as detailed case studies. The dining rituals, such as guotang and the Refectory, transform daily routines into acts of worship and practice, particularly within the palace-like dining spaces. Monastic rules and the concept of cleanliness influence the layout of monastic spaces, effectively distinguishing between sacred and secular areas. The types of food, influenced by self-sufficiency and food taboos, impact the formation of monasteries in the surrounding landscape, while the diligent labor of monks in cultivating the wilderness contributes to the sanctity of the venues. By employing anthropology as a tool for field observation and considering architectural design as a holistic mindset, this article concludes that due to the self-sufficiency of monastic lives, monks establish a sustainable agri-food space system. This ensures that food production, waste management, water utilization, food processing, and meal consumption can be sustainable practices. Food taboos are determined by the understanding of purity in both religions, leading to the establishment of a distinct spatial order for food between the sacred and secular realms. Ultimately, ordinary meals are consumed within extraordinary dining spaces, providing monks with a silent and sacred eating atmosphere. Under the overall influence of food, both monasteries have developed their own food spatial systems, and the act of dining has transformed from a daily routine to a sacred worship.

**Keywords:** monastic space; self-sufficiency; purity; agri-food space system; food spatial order; unusual dining space

## 1. Introduction

### 1.1. Study Background and Objectives

> *"Though eating is essential to continued life, both the use of food and intentional abstention from it are cultural practices revealed as the mans of expression of powerful emotions".* (Mintz 1996, p. 69)

Food holds paramount importance for human beings, and the significance of meals is universally evident across various cultures, particularly in religious contexts. Numerous anthropologists specializing in food studies have extensively researched this aspect, dedicated to exploring how we connect to the world through our food. Given that research on food is mainly approached from the perspectives of social history, ethnography, and anthropology, exploring the social and cultural significance of food, British anthropologist Goody (1982) summarized the study of food anthropology into three major tendencies: the functional approach (emphasizing the socializing functions of food), the structural approach (seeking underlying structures behind food), and cultural approaches (unearthing cultural codes within society). Representative works include "Land, Labour and Meal

in Northern Rhodesia" by Richards and Institute (1939), "The Raw and the Cooked" by Lévi-Strauss et al. (1969), and "Purity and Danger" by Douglas (1966). Peng and Xiao (2011) summarized the anthropological study of food into the following four aspects: basic functions, spiritual attributes, cultural identity, and relationship with the ecological environment.

Even though disciplines like anthropology and sociology have extensively examined various aspects of food in society, there seems to be a lack of comprehensive consideration regarding how food influences the spaces in which it is consumed, particularly in religious contexts. Scholars researching religious food tend to focus on specific aspects, such as sacrifice (Smith 1889), taboo (Frazer 1890), purity (Douglas 1966), origins (Visser 1991), and dining rituals (Shi 2020), without thoroughly exploring how these practices impact the spatial arrangements of religious sites. This limited approach hinders a holistic understanding of the relationship between food and space within religious settings. On the other hand, architectural researchers studying monastic dining spaces tend to prioritize the examination of the architecture itself (Wolfgang 1980), neglecting a broader exploration of the spatial aspects of food from an anthropological perspective. This oversight means that the influence of food on the internal dynamics of religious venues remains underexplored in architectural research.

To address these gaps, there is a need for interdisciplinary research that bridges the fields of anthropology, sociology, and architecture. By combining insights from these disciplines, one can gain a more comprehensive understanding of how food practices and beliefs shape the spatial arrangements of religious venues. This holistic approach would shed light on the intricate relationship between food and the monastic built environment, enriching our understanding of how monks interact with and respond to food within their spaces. For example, how do monks transform their approach to meals from the mundane to the spiritual? Is it through concepts, precepts, or rituals? Since a meal serves as an important component of religious life, what impact does it have on the formation of religious space? Apart from dedicated dining spaces, in what other ways does a meal influence the formation and layout of the environment for spiritual practice? This study tries to explore the spatial structure of food, specifically how its influence on religious life is reflected tangibly and intangibly in the space formation, layout organization, and even site selection of monastic venues.

Before delving into the elaboration of this research, it is necessary to provide an explanation regarding the selection of research objects, research methods, and scope.

### 1.2. Research Objects, Methods, and Scope

1.2.1. Selection of Research Objects

The selection of Han Buddhist and Cistercian monasteries as the primary research objects is based on the following three reasons:

1.  Similarity in monastic life and spatial correspondence: choosing these two as research objects is motivated by their similar monastic practices and spatial correspondences (Wang 2023). The similarities in their way of life and spatial arrangements provide a comparative basis for understanding the influence of food on the formation of religious venues.
2.  Emphasis on self-sufficient religious lives: both Han Buddhist and Cistercian monks place a high emphasis on self-sufficiency in their religious lives, particularly in essential elements such as water (Wang and Feng 2023) and food, which can be handled and cultivated by the monks themselves. The ability to achieve food self-sufficiency is crucial for the sustainability of their way of life.
3.  Comparative analysis as the analytical framework: comparative research serves as the fundamental structure of this study, allowing for the identification of universal principles. The author has already discussed this analytical approach in detail in another article (Wang 2021), and it will not be further elaborated upon here.

These reasons support the selection of Han Buddhist and Cistercian monasteries as research objects and provide a foundation for understanding the impact of food on space formation, layout organization, and site selection.

Furthermore, Guoqing Si and Poblet Monastery will serve as exemplary cases for studying the relationship between food and space. It should be noted that as important architectural cultural heritage, they have evolved continuously over a long historical period, evident in their architectural scale and specific spatial arrangements. Given that previous scholars have provided detailed records of the history and architectural evolution of both monasteries (Finestres y de Monsalvo and Guitert i Fontseré 1947; Altisent 1974; Guanding n.d.; Ding 1995), and the author has conducted in-depth analyses of their spatial layout and the ideal plans referenced (Wang 2021, 2023), this article will not repeat the discussion. On the other hand, for this study, the correspondence between life and space is the main research focus. Due to the stable influence of religious regulations, this correspondence possesses permanence, serving as a core principle that withstands the test of history. Therefore, analyses and theoretical models can be established from the perspectives of space formation, layout organization, and site selection. This spatial theoretical model will not undergo significant changes over time, especially when viewed from the perspective of human dietary habits. Despite significant technological and societal changes, human needs for food maintain a relatively simple relationship with nature.

Therefore, this study aims to establish a spatial model of food in monasteries, depicting the complete relationship between the cognition of food and the formation of monastic space and analyzing how dining is transformed from daily routine to sacred worship.

### 1.2.2. Research Methods

(1) Using anthropology as a tool for field observation

As meals hold a crucial role in religious life, adopting an anthropological approach is necessary to observe its significance throughout the entire chain. The monks' attitudes toward food influence how it is obtained, handled, and utilized in their daily lives. Furthermore, the monastic space is intricately linked to the behavioral activities of the monks. In addition to its functional aspect, the religious teachings and cultural symbolism associated with food are manifested in various monastic rituals, which, in turn, are connected to the importance of space. Out of consideration for the sacred and secular aspects of space, food also influences the monastic layout. Ultimately, food is a fundamental necessity upon which human life depends. The availability of water and food is intertwined with the monks' diligent efforts and is embedded within the potential of the surrounding environment where the monastery is located. Conversely, the selection of specific types of food required for religious practices influences the formation of the landscape surrounding the monastery. Therefore, the correspondence between space and life (time) requires anthropological involvement for observation and interpretation.

(2) Viewing architecture design with a holistic mindset

Architectural design thinking functions as an invisible hand, guiding and regulating the site selection, layout organization, and space formation in religious venues. Architecture acts as a medium that bridges the gap between the natural environment and us. Adopting an architectural design approach with a holistic mindset in research involves considering various factors such as human experiences, environmental constructions, behavioral habits, rules and regulations, religious symbolism, and construction methods. It transcends a superficial understanding of individual elements and places them within a broader context, observing them from a comprehensive spatial dimension. This holistic approach aids in understanding the overall relationships between food and space. It enables a comprehensive comprehension of the interconnections between form, function, environment, and human experience.

1.2.3. Research Scope

Through an anthropological perspective and viewing architecture design with a holistic mindset, a deep examination of the role of food in the daily and spiritual lives of monasteries leads to the development of a unique spatial model specific to monastic food practices. The objective of this study is to integrate existing foundations of food anthropology research and studies on monastic dining spaces, with the aim of proposing a research paradigm for the study of the spatial structure of food.

## 2. From Routine to Worship

*"He does not eat meat or take intoxicating drinks", "nor vegetables of the five kinds of astringent smell" [including garlic, leeks, onions]. Hence, no unpleasant smell comes about [on the breath]. He is always respected and given offerings, honoured and praised by gods and humans.* (T12n0374 2007, p. 160)

For Han Buddhists, according to The Mahayana Mahaparinirvana Sutra (大般涅槃经, written around 420–479), "*One who eats meat kills the seed of great compassion*" (T12n0374 2007, p. 52). Adhering to a vegetarian meal is considered an expression of compassion. However, not all types of vegetarian food are permissible; the consumption of the "five acrid and strong-smelling vegetables", which include garlic, asafoetida, shallot, leek, and allium, is prohibited due to their strong odor. Monks are one of the Three Jewels of Buddhism (Buddha, Dharma, and Sangha) and symbolize the embodiment of the Buddha's teachings in the world. Avoiding these strongly aromatic foods is important to maintain the appropriate image of monks.

In addition, when it comes to beverages, alcoholic drinks are strictly prohibited. This is because they have the potential to arouse sexual desires or create temptations in the hearts of monks.

*"My disciples, if you intentionally drink alcohol, there is no limit to the mistakes and violations you will make. If with your own hand you pass the wine bottle to another, you will be born without hands for five hundred lifetimes—how much worse if you drink the wine yourself? You should not encourage any person to drink, nor any sentient being to do so; how much worse if you yourself drink alcohol yourself? If you intentionally drink, or encourage someone else to do so, you have committed a minor transgression of the precepts".* (T1484 2017, pp. 48–49)

Very similar to Han Buddhists, according to the Rule of St. Benedict—set by an Italian abbot, Benedict of Nursia, in the sixth century—while still fundamental in monks' daily life, Cistercian monks also adhere to a simple lifestyle to prevent their hearts from becoming "*weighted down with surfeiting*" (Clarke and Society for Promoting Christian Knowledge 1931).

"*And let all abstain entirely from the eating of the flesh of quadrupeds, altogether excepting from this rule the weak and the sick … neither surfeiting nor drunkenness creep in*" (Clarke and Society for Promoting Christian Knowledge 1931). Therefore, vegetarian food becomes their primary choice to restrain their desires. However, for weak or sick monks, meat is allowed as a means to supplement nutrition and aid in their recovery. "*let the eating of flesh meat be conceded to the sick and especially to those who are weak, for their recuperation; but when they shall have got better let them all abstain from flesh meat as usual*" (Clarke and Society for Promoting Christian Knowledge 1931). Although monks were permitted to drink wine, moderation was always emphasized. It was important not to overindulge or misuse alcohol. Because "*Wine makes even the wise to fall away*" (Clarke and Society for Promoting Christian Knowledge 1931).

In addition to the regulations on food, there are also detailed guidelines about at what hours the brethren ought to have their meals. These guidelines regarding mealtimes align with the principles of working at sunrise and resting at sunset, reflecting the agricultural society of the time. The Rule of St. Benedict specifies, "*From holy Easter until Pentecost, let the brethren dine at the sixth hour and sup about sunset … And indeed on all occasions let the hour,*

*whether of supper or dinner, be so suitably arranged that everything be done by daylight*" (Clarke and Society for Promoting Christian Knowledge 1931).

Based on the regulations formulated by both religions, the food choices they make have several common characteristics: (1) Self-Sufficiency: both religions emphasize a self-sufficient lifestyle, where monks rely on their own labor to obtain food relatively easily. (2) Clean and Healthful: the selected food is clean and healthful, ensuring that it has no negative impact on the body and the image of the monks. (3) Moderation: there is an emphasis on the quantification of meals and avoiding excessive eating, which helps monks maintain self-discipline and reduces the likelihood of succumbing to lustful desires. (4) Mealtime Routine: the timing of meals is set as a routine closely related to the daily work schedule of the monks. This structured approach to mealtime ensures a disciplined lifestyle and facilitates the synchronization of their spiritual practices and daily responsibilities.

Overall, these commonalities in food regulations reflect the shared values of simplicity, discipline, and mindfulness in the monastic lives of both Han Buddhist and Cistercian monks. In addition to regulations regarding food, ensuring a solemn and orderly dining process is also of great importance in both religions. Both Han Buddhist and Cistercian traditions have specific guidelines and procedures for their dining rituals, which are closely related to the characteristics of the dining spaces themselves. The cases of the Zhaitang of Guoqing Si and the Refectory of the Poblet Monastery are going to be discussed below.

*2.1. Ritual and Worship Space*

2.1.1. Zhaitang and Refectory

> *"Thus fountains became receptacles of living water, dorters became chambers of sleep, chapter-houses enshrined the gravity and solemnity of chapter-sittings, and refectories the importance ascribed to the common meal in the regimen of ascetics. The meagre fare was eaten in princely dining-halls, which sometimes rivalled churches in their size and magnificence".* (Wolfgang 1980, p. 97).

Although both Han Buddhist and Cistercian monasteries adopt rectangular floor plans for their dining spaces, similar to their worship spaces (Wang 2021), there are differences in the entrance approach between them. In Han Buddhist monasteries, the emphasis is on the horizontal extension of the space, where one enters from the long side of the building. On the other hand, in Cistercian monasteries, the entrance is from the short side, emphasizing the vertical depth of the space. In terms of architectural composition, both types of buildings can be categorized into two types: standalone structures (Figures 1 and 2) or buildings with functional spaces on the upper levels (Figures 3 and 4). While standalone buildings and composite structures may differ in the perception of internal space ambiance, their floor plans exhibit similarities due to the pursuit of practicality and operational ritual requirements.

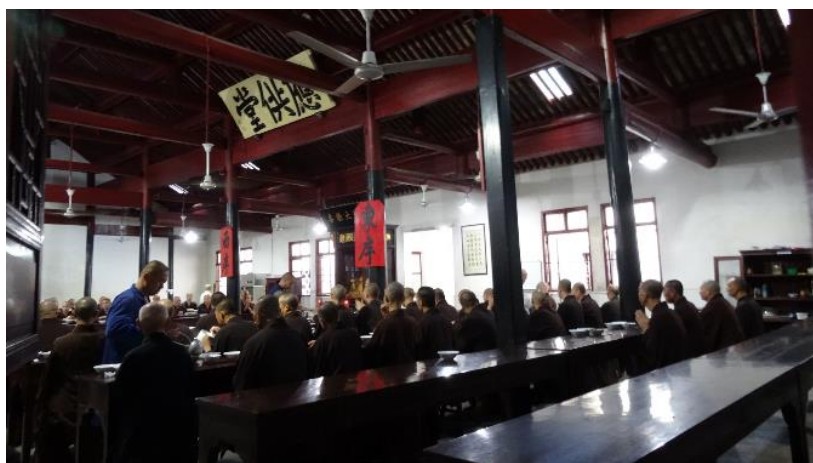

**Figure 1.** Zhaitang as standalone building in Tiantong Si. Photo by the author.

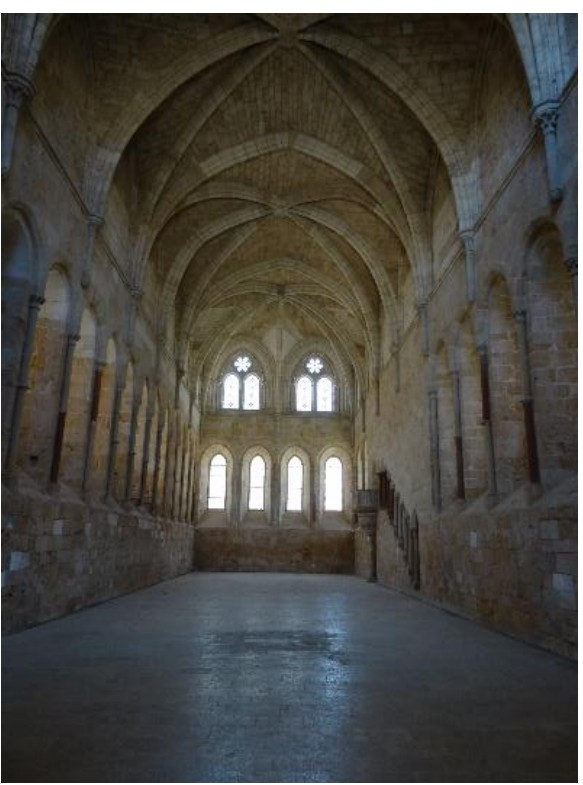

**Figure 2.** Refectory as standalone building in the Monastery of Huerta. Photo by the author.

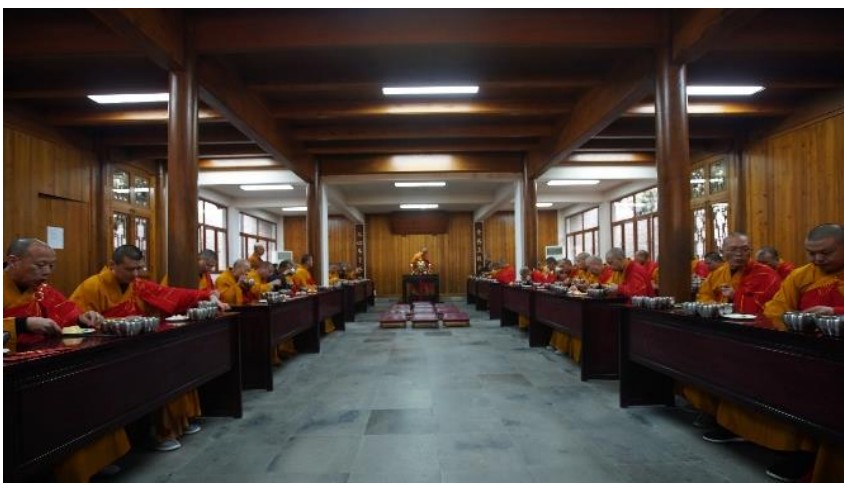

**Figure 3.** Zhaitang with upper floor in Qita Si. Photo by the author.

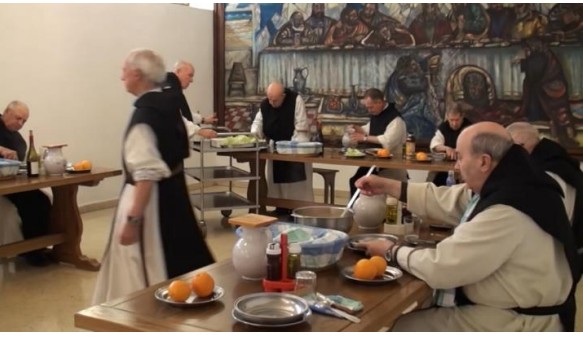

**Figure 4.** Refectory with upper floor in Oliva Monastery. Image cited from Monasterio de la Oliva, https://www.monasteriodelaoliva.org/nuestra-vida/. Accessed on 17 July 2023.

In Guoqing Si, the monks' dining hall is called Juxian Ge (聚贤阁), which translates to "Hall of Gathering Virtuous Beings". This name signifies that the hall is meant to host all the saints or virtuous individuals. In other monasteries, a similar dining hall may be referred to as Wuguan Tang (五观堂), which means the Hall of the Five Contemplations. It refers to a specific hall or space in a monastery where monks gather to recite and practice the Five Contemplations before and after meals. Or it may be referred to as the Zhaitang (斋堂), which means the place for vegetarian meals in the monastery. Zhai, according to the research by Yan (2007, pp. 296–310), can be referred to as vegetarian meals, dining rituals, practice precepts, and Buddhist vegetarian feasts. In this article, the dining space in Han Buddhist monasteries is uniformly referred to as the Zhaitang.

The Zhaitang is situated on the first east axis of Guoqing Si, perpendicular to the central axis where the main temples for Buddha worship are located (Figure 5). It has two floors and five bays (Figure 6). The first floor serves as a dining hall for monks, while the second floor is used for storage, known as Kufang (库房). It measures 24 m in length and 11 m in width, with three doors on the south wall (Figure 7). The interior layout is closely related to the rituals of dining. Monks are divided into two groups, called Dongxu (东序) and Xixu (西序), sitting and eating face to face (Figure 8). The Abbot sits in the central place, in front of which there is a Buddha statue with six futons for pilgrims to offer their adorations (Figure 9). In the south of the Zhaitang, there is a wide corridor with two large stone basins for monks to wash their hands and clean their mouths before and after meals (Figure 10). The spacious courtyard in the south also serves as a space for monks to walk around after meals and is also used for drying rice.

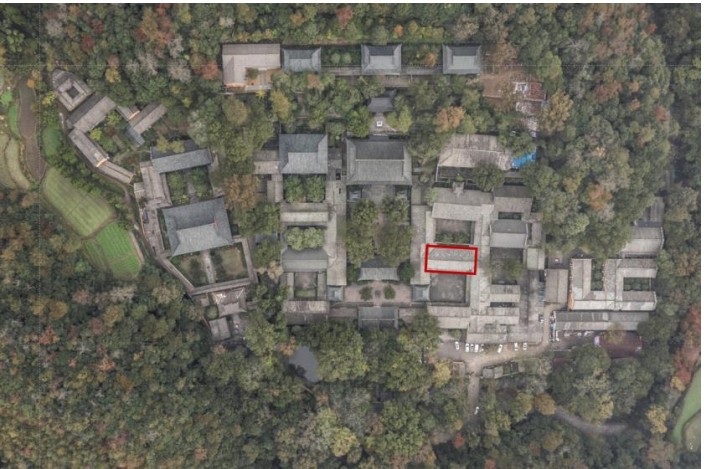

**Figure 5.** Aerial view of Guoqing Si, with the red box indicating the Zhaitang. Photo by the author.

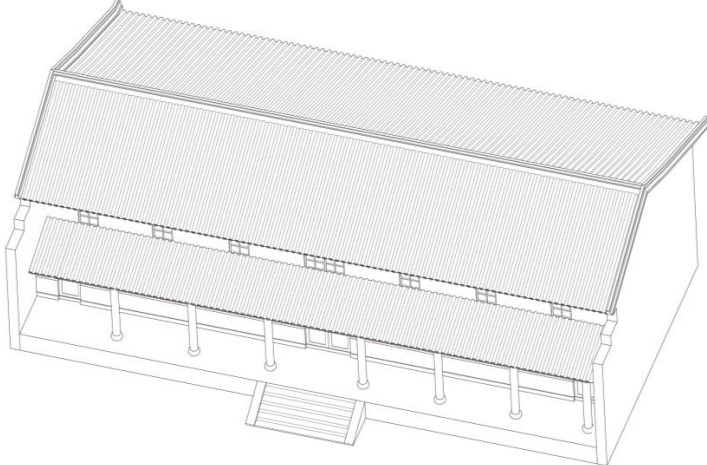

**Figure 6.** Axonometric view of Zhaitang. Drawn by the author.

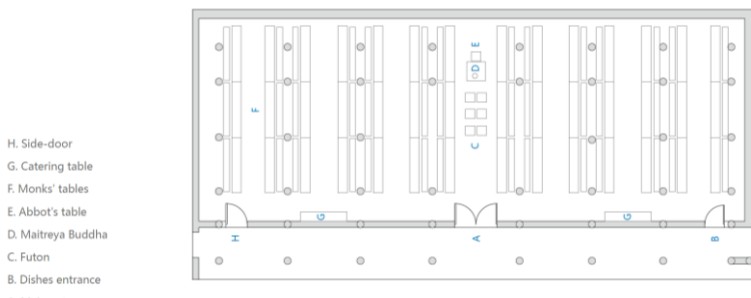

H. Side-door
G. Catering table
F. Monks' tables
E. Abbot's table
D. Maitreya Buddha
C. Futon
B. Dishes entrance
A. Main entrance

**Figure 7.** Plan of the Zhaitang. Drawn by the author.

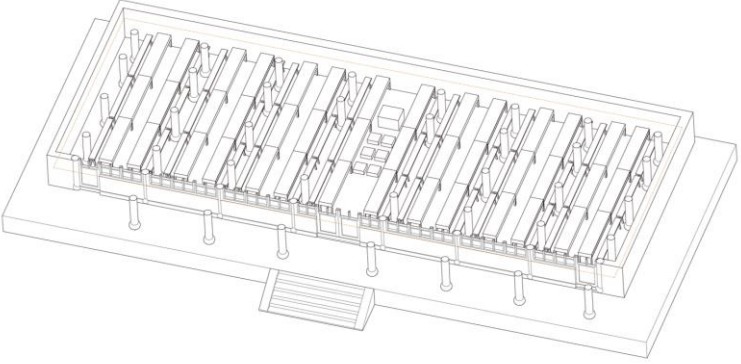

**Figure 8.** Axonometric section of the Zhaitang. Drawn by the author.

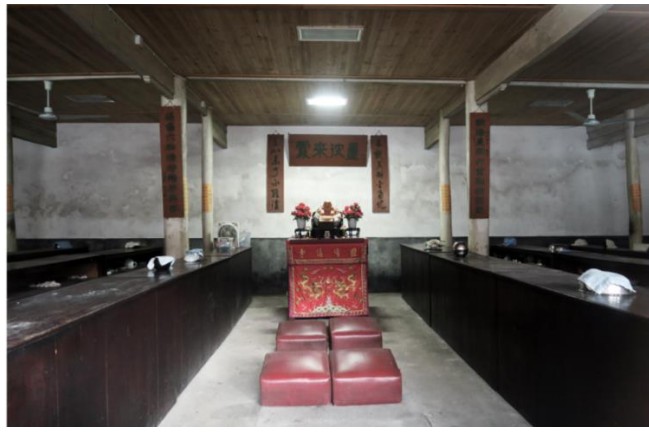

**Figure 9.** The interior of Zhaitang. Photo by the author.

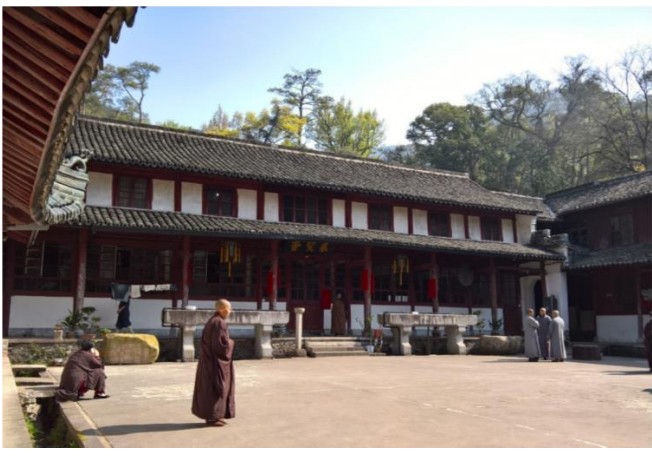

**Figure 10.** The courtyard in front of Zhaitang. Photo by the author.

The Refectory is located on the north gallery of the Cloister of Poblet Monastery (Figure 11). Its floor is rectangular, measuring 36 m in length by 12 m in width (Figure 12). Apart from the church, the Refectory is the most splendid building in the Cloister (Figure 13). The great chamber is covered by a slightly pointed stone barrel vault (Figure 14), indicating its original construction date around the 12th century, close to the Romanesque style. Three sturdy ribbed arches are sustained by plain capitals, which leave a spacious and pillar-free interior. Just in front of the Refectory, there is a fountain pavilion constructed at the same time as the Refectory (Figure 15), where monks used to wash their hands before entering the Refectory after working in the fields. The sense of spaciousness in the interior is further emphasized by placing monks' dining tables along the three walls. It is understandable that the abbot's seat is once again located in the center of the north wall, under the Cross of Jesus, directly opposite the entrance to the Refectory. A pulpit is built into the thick east wall, where a monk on duty can access the high platform through a narrow stair hidden in the wall and read sacred books, history, hagiographies (lives of saints), and biographies, while the rest of the community eats in silence (Figure 16).

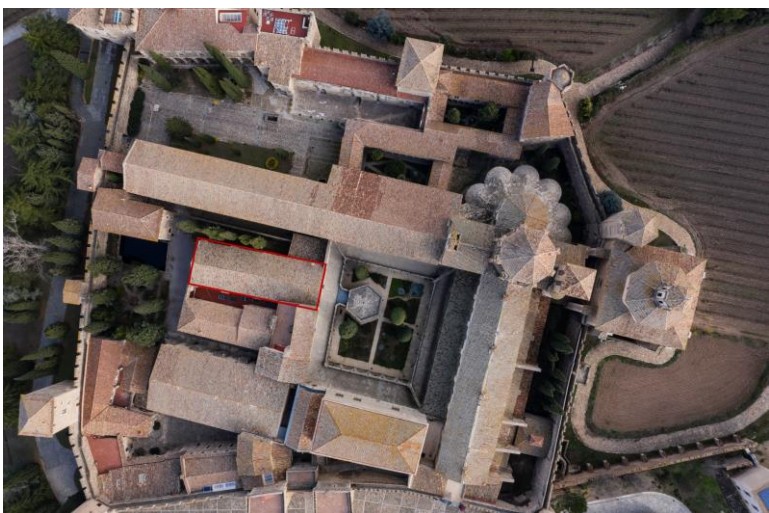

**Figure 11.** Aerial view of Poblet Monastery, with the red box indicating the Refectory. Photo by the author.

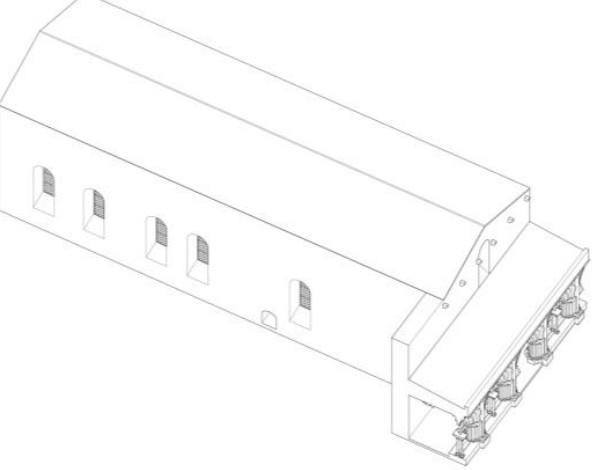

**Figure 12.** Axonometric view of the Refectory. Drawn by the author.

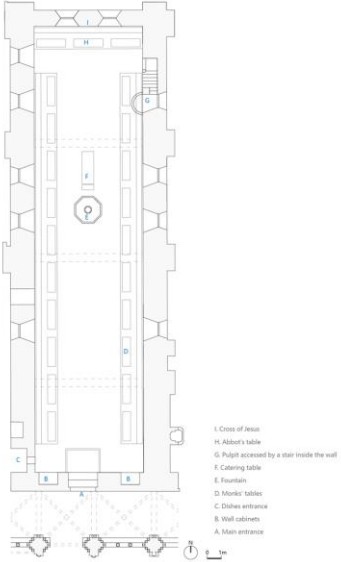

**Figure 13.** Plan of the Refectory. Drawn by the author.

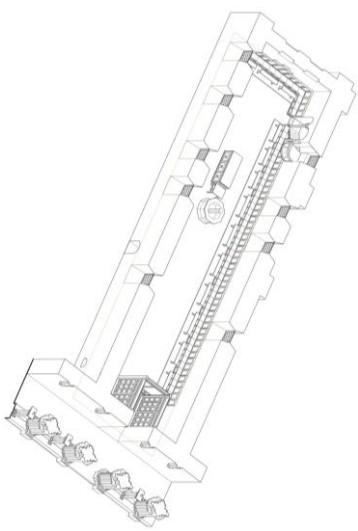

**Figure 14.** Axonometric section of the Refectory. Drawn by the author.

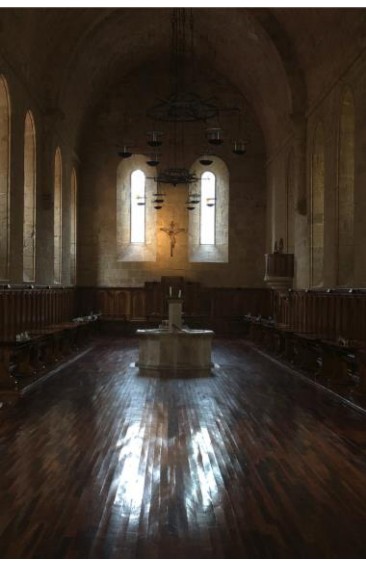

**Figure 15.** Interior setting of the Refectory. Photo by the author.

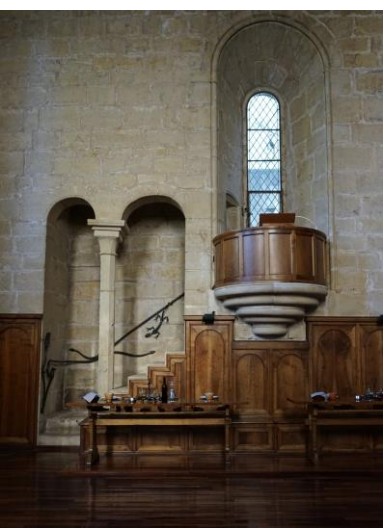

**Figure 16.** Pulpit of the Refectory. Photo by the author.

Both the Zhaitang and the Refectory adopt the dining form of a divided meal system, with tables placed symmetrically in two rows. This ritual act of dividing the meal system is in line with the spatial form of the building, which is symmetrical in the central axis and accentuates the centrality of the building. In the center are the abbot and the prior, and in front or above them are the statues of Maitreya and Jesus on the Cross. Clearly, in a space like the dining hall, the distinction between the sacred and secular order is also evident, with the sacred elements occupying the central and highest positions. Next in importance are the abbot and the monks on both sides.

In China, the divided meal system is also a deeply traditional practice. An example can be found in the Main Chamber of Cave 61 in the Mogao Grottoes, where the lower section of the mural from the Five Dynasties period (A.D. 907–960) depicts fifteen panels illustrating Buddhist stories of secular life. The thirteenth panel, "The Marriage of the Crown Prince" (Figure 17), portrays a feast for guests, with two rows of people sitting facing each other and the host seated in the middle. For Cistercians, we can observe a similar arrangement in the printmaking of the Refectory at Port-Royal des Champs (Figure 18), where nuns are shown having their meals in the Refectory, seated in two rows.

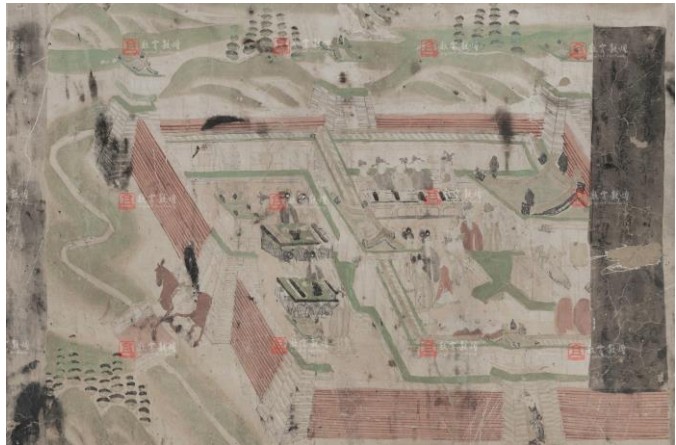

**Figure 17.** The Marriage of the Crown Prince in the Cave 61 in the Mogao Grottoes. Cited from Digital Dunhuang, https://www.e-dunhuang.com/cave/10.0001/0001.0001.0061. Accessed on 17 July 2023.

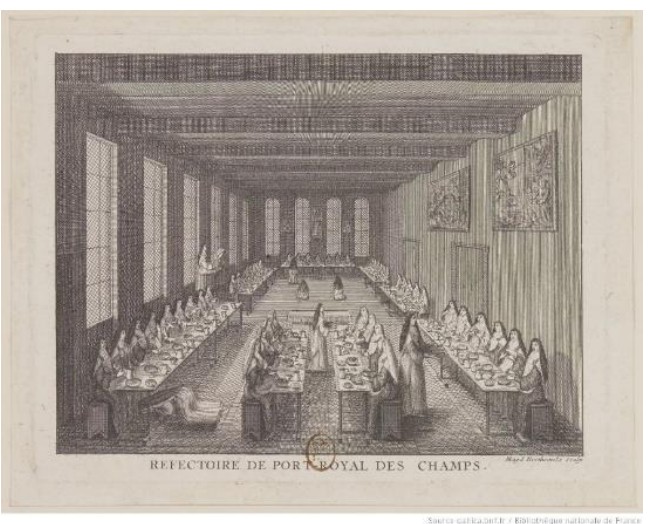

**Figure 18.** Horthemels, Louise-Magdeleine (1686–1767). Réfectoire de Port-Royal des Champs [Image fixe]: [estampe]/Magd Hortemels sculp. (1709–1710). Cited from https://catalogue.bnf.fr/ark:/12148/cb44485244r. Accessed on 17 July 2023.

The advantage of the divided meal system is that it keeps distance among monks and avoids unnecessary conversations and physical contact. Simultaneously, it emphasizes the sacredness of the central space. The Refectory is intended to be a sacred and solemn space, not meant for traditional lively feasts or gatherings. Instead, it is more like the interior arrangement of a court (Figure 19), resembling a solemn parliament where monks listen to the voice of the Lord, express gratitude, reflect, and repent. The monks sit against two walls, leaving an empty space in the middle. In contrast, during gatherings, the food occupies the central position (Figure 20), surrounded by people, with pathways for passage and service. In fact, only during specific rituals, such as offering to the heavens (Figure 21) or the distribution of the Eucharist (Figure 22), does the food offered to the Buddha and the food representing the flesh and blood of God occupy the central position. It is often the peak moment of food consumption in the monasteries during these occasions.

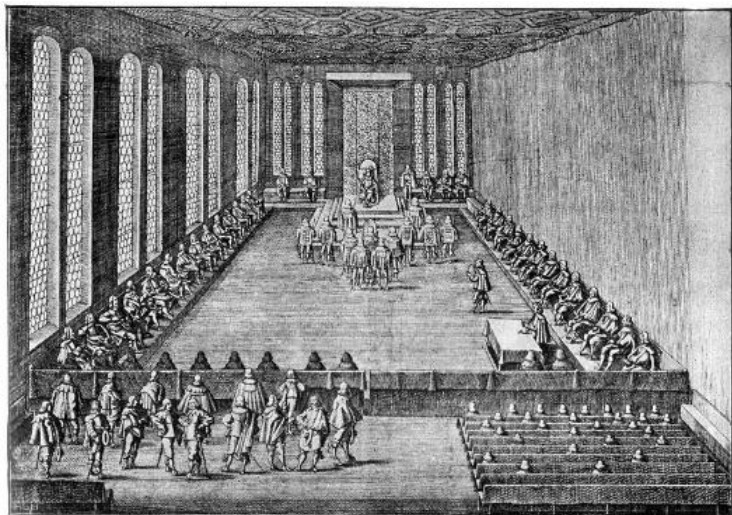

**Figure 19.** Meeting of the Perpetual Imperial Diet in Regensburg in 1640, after an engraving by Matthäus Merian. https://en.wikipedia.org/wiki/Imperial_Count. Accessed on 17 July 2023.

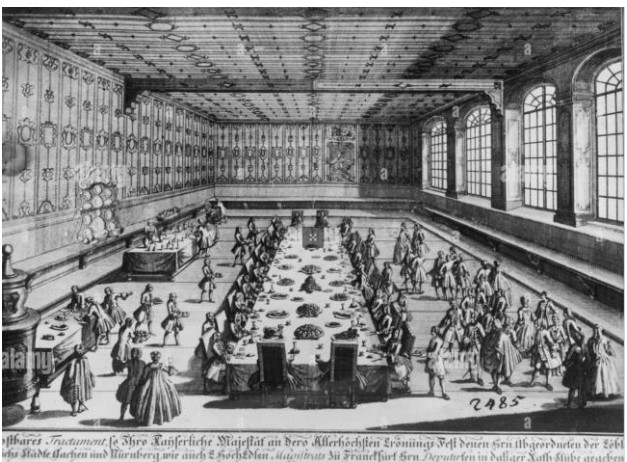

**Figure 20.** Gala dinner on the occasion of a coronation feast in the historic Roemer building in Frankfurt am Main. Cited from https://www.abebooks.com/art-prints/%C2%84Kostbares-Tractament-welches-Ihro-R%C3%B6m-Kayserl/31539509644/bd. Accessed on 17 July 2023.

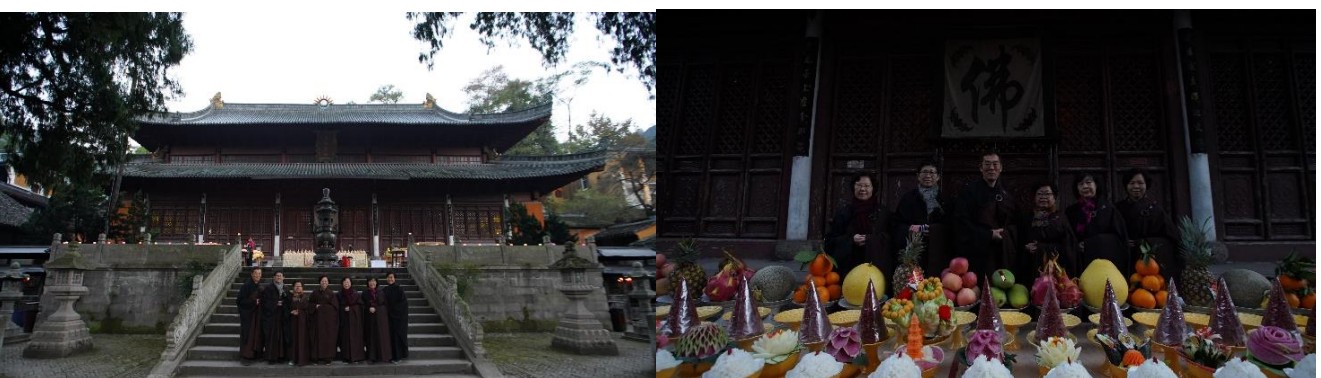

**Figure 21.** Offering to the heavens in Guoqing Si. Photo by the author.

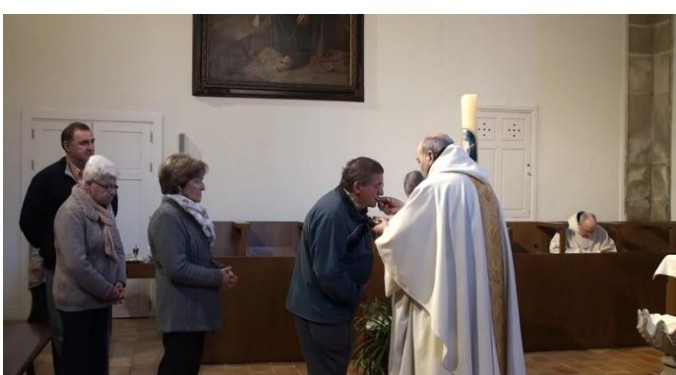

**Figure 22.** Distribution of the Eucharist in the Oliva Monastery. Cited from the Monastery of Oliva, https://www.monasteriodelaoliva.org/nuestra-vida/. Accessed on 17 July 2023.

### 2.1.2. Guotang and Refectory

Both in Han Buddhist and Cistercian monasteries, dining spaces are exclusively dedicated and independent, and their importance in terms of space is second only to worship space. Although the food is usual, the way of obtaining a meal is an unusual event in the monk's daily life. Just as Wolfgang (1980) said, "*meagre fare was eaten in princely dinninghalls*". Therefore, in both monasteries, the dining hall serves as another crucial place to realize religious practice instead of being merely a canteen.

> *"The dining ritual is not only a meal ceremony but also a Dharma event. Firstly, there are offerings to the Buddha and offerings to all sentient beings, followed by the dining of monks. The entire process appears very solemn and serene".* (Shi 2020, p. 86)

In Han Buddhist monasteries, the process of monks' dining is referred to as "guotang" (过堂). The term "guotang" encompasses the entire process of monks entering the dining hall, taking their seats, eating, and leaving after finishing their meals (Figure 23). In detail, when the wooden Bang (big wooden fish) and the cloud board (cloud-shaped iron plate) hanging in front of the Zhaitang are struck, monks lined up in two rows and entered the Zhaitang. If there are pilgrims, then monks have to walk in front, and pilgrims follow behind. Further, male and female pilgrims have to be separated. Once entering the Zhaitang, monks sit on two sides, while the abbot sits in the center, behind the statue of Maitreya Buddha. In the same way that traditional Chinese rituals begin with offering sacrifices and then proceeding to the feast, from the contents of the Ritual of Midday and Noon Meals (二時臨齋儀), it can be understood that before offering the morning and noon meals, the first step is to make offerings to the Buddhas and Bodhisattvas. Then, food is offered to sentient beings in the hungry ghost realm. Following that, offerings are made to the Sangha (the community of monks and nuns) and other practitioners. Finally, at the conclusion of the ritual, the merits generated from the offerings are dedicated through the Dharma (teachings) to those who have provided alms and food as acts of generosity.

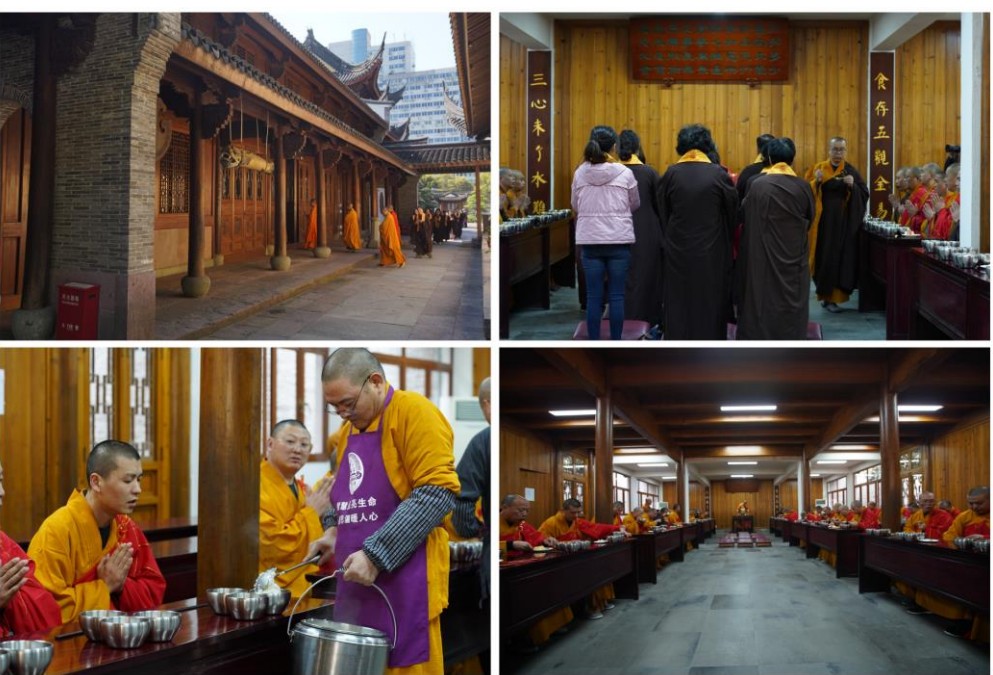

**Figure 23.** Process of guotang in the Buddhist monastery. Photo by the author.

During the dining, Wuguan is closely related to monastic religious discipline, which means monks have to practice the Five Contemplations during eating. In short, monks have to be grateful for the hard-won food, reflect on one's own merits; one's virtue and cultivation; whether one can bear the food and treat the food equally; prevent the three thoughts of greed, hatred, and ignorance; and have righteous thoughts about the food. Usually, food is considered good medicine for curing physical hunger, and eating helps to prolong life and is good for practice.[1] Generally, there is a couplet in the Zhaitang, "If you have Dharma in your heart when you eat, even gold can be digested; otherwise, a drop of water will be difficult to swallow (五观若明金易化, 三心未了水难消)".

Monks must uphold righteous thoughts when eating so that distracting thought will not dominate their hearts. For them, the symbolic meaning of food has exceeded its deliciousness. Corresponding to this is a set of rigorous rituals. The signal to eat is sent by the

instrument. For Buddhist monks, eating is a simple and quick issue that usually lasts less than 15 min, which means the meal is quite simple (Figure 24). During the dining, if monks have any requirement for food, they have to express their needs through gestures instead of speaking (Figure 25). Once the meal is over, monks have to recite again and walk in two rows to leave the Zhaitang. According to Buddhist precepts, there are only two meals a day, in the morning and at noon, and no more food is consumed after noon. Based on this, Han Buddhists also adhere to the requirements of vegetarianism. However, due to the self-sustaining life in the monastery, some monks who work as laborers still need a meal at night to supplement their physical strength. This dinner is called "medicinal food" and is eaten purely to maintain good health.

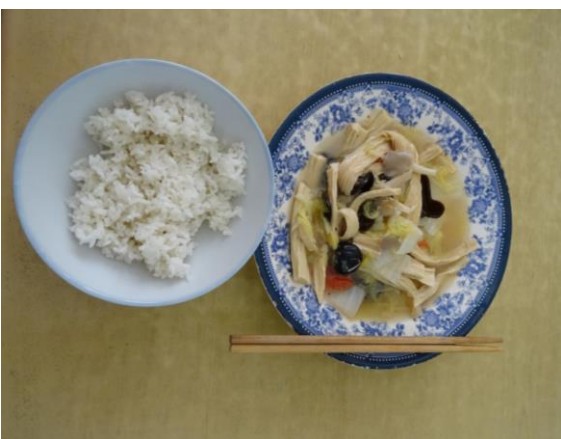

**Figure 24.** Simple vegetarian meal. Photo by the author.

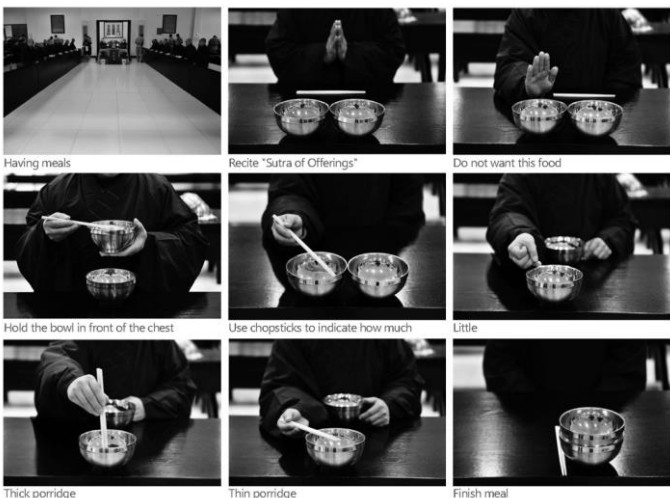

**Figure 25.** Gestures to express food requests in Han Buddhist monasteries. Cited from the journal of Ren Shi Jian 人世间, by Monk Yunxiang. http://www.oliannews.com/xsjl/2018/09-21/244199.shtml. Accessed on 17 July 2023.

> *"Let the deepest silence be maintained that no whispering or voice be heard except that of the reader alone. But let the brethren so help each other to what is needed for eating and drinking, that no one need ask for anything. If, however, anything should be wanted, let it be asked for by means of a sign of any kind rather than a sound. And that no one presume to ask any questions there, either about the book or anything else, in order that no cause to speak be given [to the devil] (Eph 4:27; 1 Tm 5:14), unless, perchance, the Superior wisheth to say a few words for edification".* (Clarke and Society for Promoting Christian Knowledge 1931)

In Cistercian monasteries, the process of monks eating is referred to as "refectory". It is a communal meal that monks eat in silence and humility, during which a monk reads from the Bible or other spiritual texts. Monks are not allowed to talk during the meal since this practice is meant to encourage self-reflection and a deeper connection with God. The meals served in the Refectory are typically simple and vegetarian, reflecting the monks' commitment to living a life of simplicity and austerity. The Refectory is not only a time for nourishing the body but also the soul, as the monks strive to deepen their spiritual connection while partaking in their meals. Vegetarian food (Figure 26) is preferred by Cistercian monks, except "the very weak and the sick abstain altogether from eating the flesh of four-footed animals". Monks' daily life is full of gravity and rituals, and eating is no exception. The Rule of St. Benedict regulates the quantity of food and drink that a monk should take and sets the rule in detail on how to proceed with the process of eating.

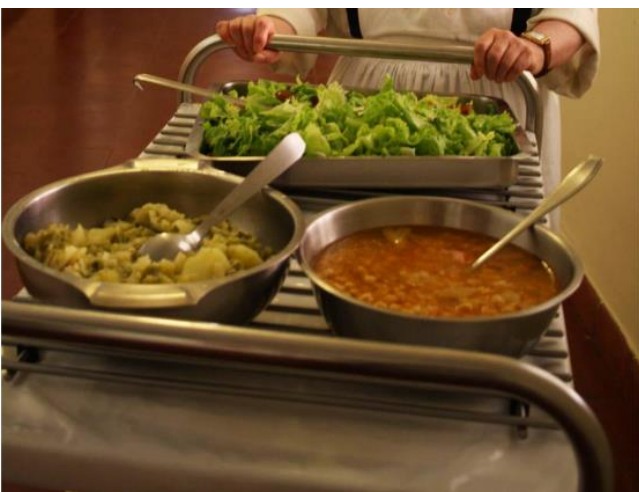

**Figure 26.** Types of meals in Cistercian monasteries.

In the Poblet Monastery, breakfast is served buffet-style, allowing monks to choose the food they want. The meal typically includes soup, muesli, bread, fruit, cheese, coffee, milk, yogurt, and other items. After breakfast, everyone washes their own dishes. Breakfast is simple and typically lasts for 15 minutes, from 8:45 to 9:00. The Abbot gives a brief speech, mainly for the blessing of the food, and then rings a bell on the table to signal the start of the meal. Lunch is a more formal affair and lasts for half an hour, from 13:15 to 13:45. Monks take turns working in the kitchen, with two monks preparing the food, two serving the plates, and one reading from a spiritual text. After lunch, another two monks wash the dishes. Before lunch, the Abbot gives a short speech of around two minutes, mainly for the blessing of the food. Then, a monk on duty can access the high platform through a narrow stair hidden in the wall and read for around five minutes from a religious text or a book about Christian life. The rest of the monks sit in their assigned seats, waiting for the food to be served, listening, and eating. When they finish lunch, the Abbot gives thanks, and the monks raise their voices in agreement by saying "Amen". Lunch usually consists of three courses, with the first being soup, macaroni, salad, or potatoes; the second being fish or meat; and the third being dessert, fruit, or flan. At each table, there is a bottle of wine for three monks, although they typically only drink a small amount. Dinner is similar to lunch and also lasts for half an hour, from 20:30 to 21:00. However, during the entire mealtime, monks are not allowed to speak. If they need anything regarding their food, they express it through gestures (Figure 27).

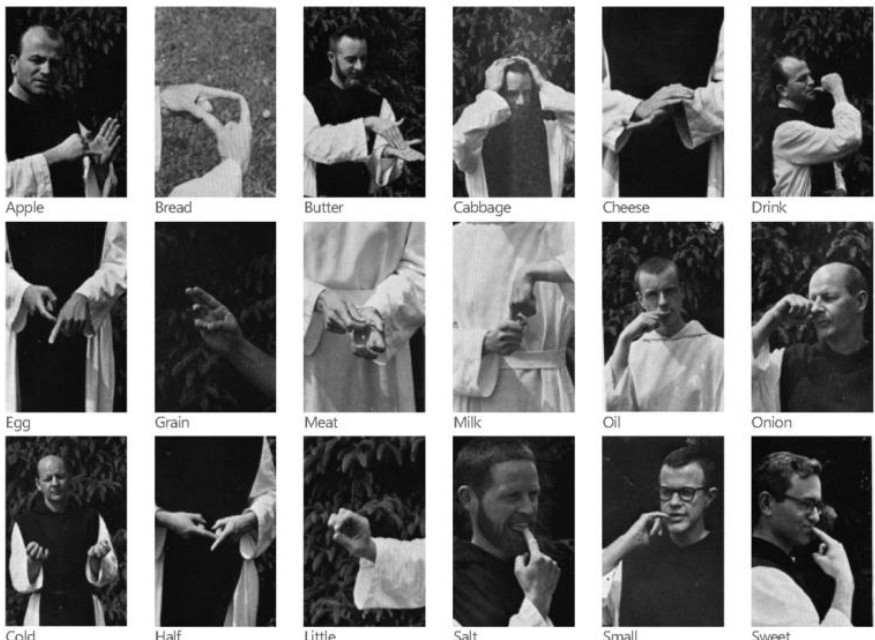

**Figure 27.** Gestures to express food requests in Cistercian monasteries. Cited from the book *The Cistercian sign language: A study in non-verbal communication* by Robert A. Barakat (1975).

Both for Han Buddhist and Cistercian monks, the regulation of dining serves the purpose of emphasizing the internal order of the religious community. Group living requires the establishment of order to ensure effective organization. We can imagine that in a monastery where more than 100 monks live, if everyone eats and talks freely, then the mealtime is as chaotic as a vegetable market. The rituals effectively control the pace of dining, with the monks starting and ending their meals at the same time. The detailed regulations regarding food also indicate that the monks do not have the freedom to choose food arbitrarily. Instead, they can only select the type and quantity of food within the prescribed guidelines. Therefore, the whole process of dining is a regulated and ritualistic one considered part of their spiritual practice.

### 2.1.3. Daily Routine and Daylight

Even though the Zhaitang and the Refectory have the same function, the way they organize the interior layout is quite different. The light of the Zhaitang only comes from the south facade, so in this scenario, the Buddha statue is fully illuminated from the front, while the monks are illuminated from the side. This is similar to the Mahavira Hall, where the scale of the Buddha and humans are distinguished through the organization of lighting. In the Zhaitang, the southern window fully illuminates the space. The height of the window is higher than the seated eye level, preventing monks from looking outside or being disturbed by passersby.

> *"The Cistercians made an innovation in the placing of their refectory. It was built at right angles to the cloister, probably less for the commonly asserted reason of giving it more light, than to leave room for a kitchen between the refectory and the house of the conversi".* (Wolfgang 1980, p. 76)

Windows on the façade of Cistercian monasteries are also attached with importance. This is not only for aesthetics but also for lighting the interior for monks to proceed with the rituals of the Refectory: "*they will not need lamp-light during their meal; but let everything be finished whilst it is still day. But at all times let the hour of meals, whether for dinner or for supper, be so arranged that everything is done by daylight*" (Clarke and Society for Promoting Christian Knowledge 1931). Therefore, the Refectory is generally set in the north–south orientation and arranged perpendicular to the church, which satisfies the need for lighting and avoids

the impact of noise and smell on the church as much as possible. In the Refectory, there are high windows on all four walls to ensure that everyone's face is illuminated. Light is considered a symbol of God, under which monks receive their food. It is worth noting that the integrity of the space and the division of sacredness and secularity are again limited by the position of the window.

In both cases, the windows are higher than the monks' heads, suggesting the space of divinity (Figures 28 and 29). Monks are not expected to communicate with the exterior through the windows. In other words, the windows in both cases serve more for illumination and ventilation, allowing monks to concentrate on their dining rituals without being distracted.

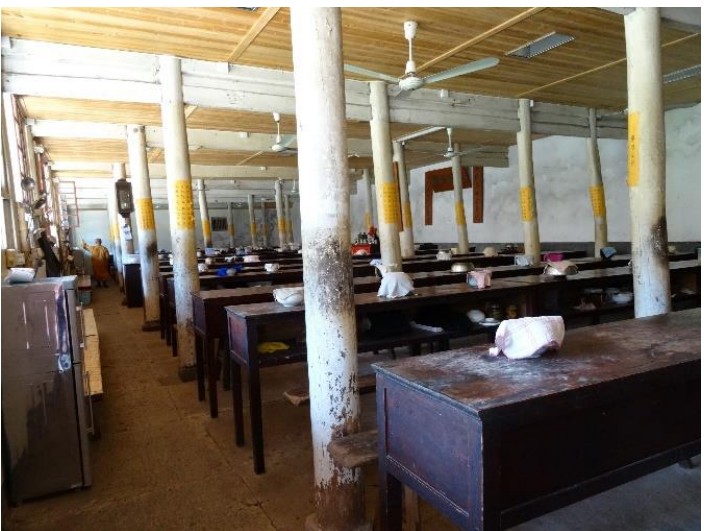

**Figure 28.** Windows are higher than the seated eye level in the Zhaitang. Photo by the author.

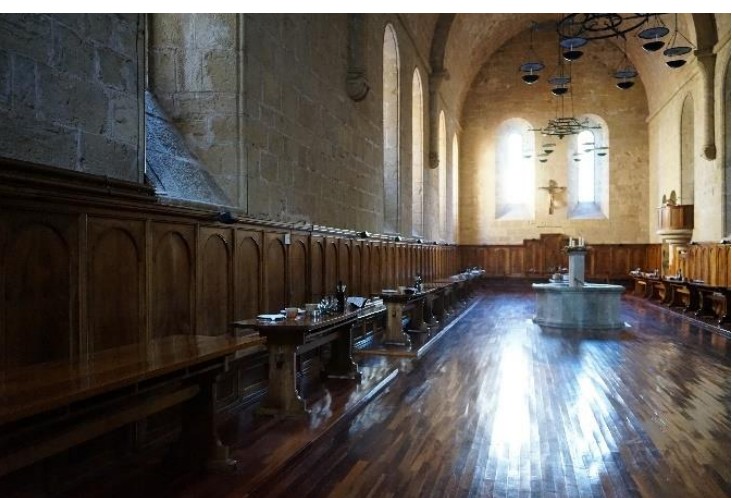

**Figure 29.** Windows are higher than the seated eye level in the Refectory. Photo by the author.

In the ideal plan (Figure 30), the earliest surviving plan of a complete Benedictine monastery drawn in 816, the Refectory is parallel to the church, and above it is the vestiary (cloister), closely connected to the dormitory (Wang 2023). However, in later evolution, the Refectory gradually became the tallest building in the monastery, second only to the church. It became a large, single-story space, further emphasizing that it is a solemn, dignified, and serene space.

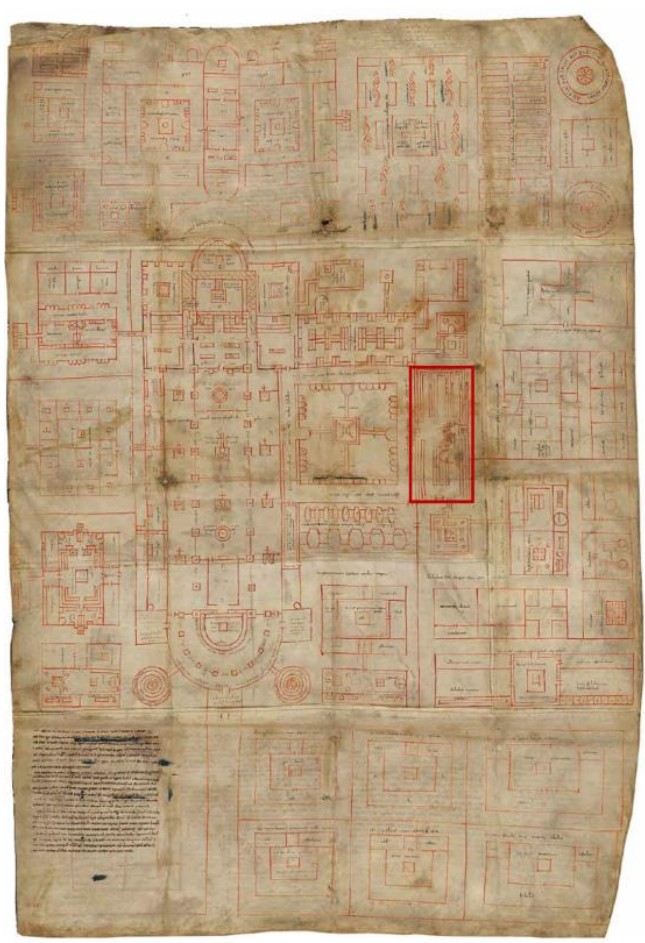

**Figure 30.** Plan of St. Gall, with the red box indicating the refectory. Image cited from the collection of St. Gall Manuscripts, UCLA Library Digital Collections with title St. Gallen, Stiftsbibliothek, Codex 1092, https://digital.library.ucla.edu/catalog/ark:/21198/zz002kp2b7. Accessed on 17 July 2023.

In both Han Buddhist and Cistercian monasteries, monks strive for a self-sufficient lifestyle that aligns with their spiritual practice. Such pursuit leads them to select food that is suitable for their monastic life and reinforce it as a daily routine through rules, scriptures, and disciplinary regulations. They typically opt for vegetarian meals to minimize harm to living beings and cultivate compassion. The selection of food is guided by the principles of non-violence and non-attachment to worldly desires. To ensure adherence to their practices, monks follow strict rules and regulations governing their meals and establish a dining routine that finally supports them in both nourishment and spirituality. This intentional approach to food helps monks maintain focus on their spiritual path and reinforces their commitment to a self-sufficient way of life. "*Regulation of meals entailed appropriate architecture*" (Wolfgang 1980, p. 79). The dining spaces in both monasteries are not treated as casual areas; rather, they are carefully designed with proper layouts and a sacred atmosphere. "*Religious piety and beliefs are not only the source of Western morality but also the force that supports Western behavioral norms*" (Fei 1992, p. 72). Prescribed ceremonies and daily ceremonial education play a significant role in cultivating a sense of awe and conviction among monks. The combination of simple and sober food, an austere and aesthetic space, and a solemn and silent dining process ensures that monks partake in their meals in a structured and disciplined manner. These daily ceremonial practices become an integral part of their spiritual journey, providing opportunities for reflection, gratitude, and the deepening of their connection to their faith.

*2.2. Purity and Food Layout*

2.2.1. Convenience and Safety

Eating is a daily routine for monks, and work related to eating has to be efficient and convenient. The divided meal system is adopted in both dining halls. In both buildings, monks enter through the central door while servants of the kitchen enter through the side door. After monks sit down, a few monks on duty in the kitchen are responsible for distributing food. To maintain the holiness of the dining, itineraries have to be well coordinated so that monks and the servants of the kitchen will not interrupt each other.

In Guoqing Si, the large old kitchen is located in the eastern part of the Zhaitang's courtyard, next to the Monks' Dining Hall and Workers' Dining Hall (Figure 31). In order to prepare meals for more than 100 monks, a spacious courtyard is necessary for washing and preparing vegetables (Figure 32). Guoqing Si still uses traditional firewood to cook meals (Figure 33), and the firewood comes from the mountains where the monastery is located. Once the food is cooked, it is ladled into wooden barrels and carried into the Zhaitang through the east side door, while monks enter the Zhaitang through the central door.

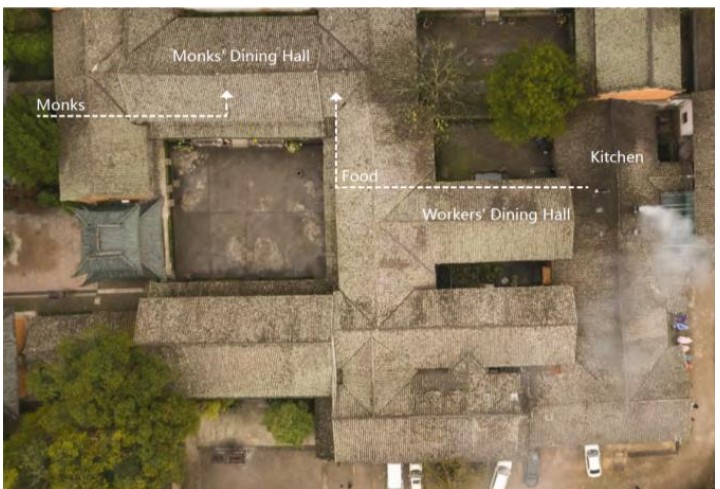

**Figure 31.** Aerial view of the Zhaitang of Guoqing Si. Photo by the author.

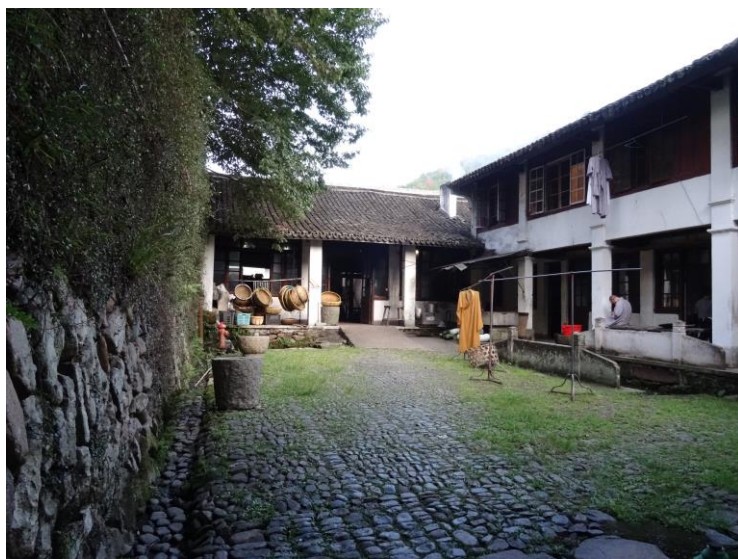

**Figure 32.** Kitchen courtyard in Guoqing Si. Photo by the author.

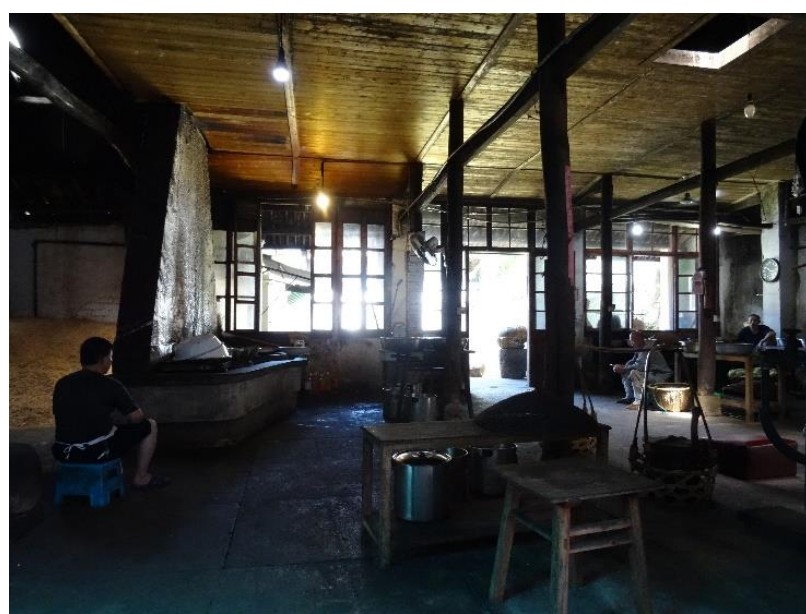

**Figure 33.** Old kitchen in Guoqing Si. Photo by the author.

The Cistercian Refectory is typically located perpendicular to the cloister, which is different from the Benedictine layout. This positioning offers several advantages. Firstly, only one kitchen needs to be constructed between the Refectory of the lay brothers and that of the monks (Figure 34), which ensures that both refectories can maintain the same dining rituals. Additionally, this layout is more flexible, as it allows for the Refectory to be easily expanded or rebuilt to accommodate a growing community. Keeping the kitchen, pantries, and warming room nearby is also more practical. Furthermore, this layout provides better lighting as windows can be placed on two or three sides. To ensure that the monks are served their food in a timely manner, especially during the cold winter months, a hole is typically pierced in the south wall of the Refectory near the main entrance (Figure 35). Today, the old kitchen is no longer in use and is only open to tourists. A wooden door (Figure 36) has been added to the center of the south wall to connect the Refectory to the new kitchen (Figure 37). This arrangement for passing food also helps to maintain the silence of the cloister.

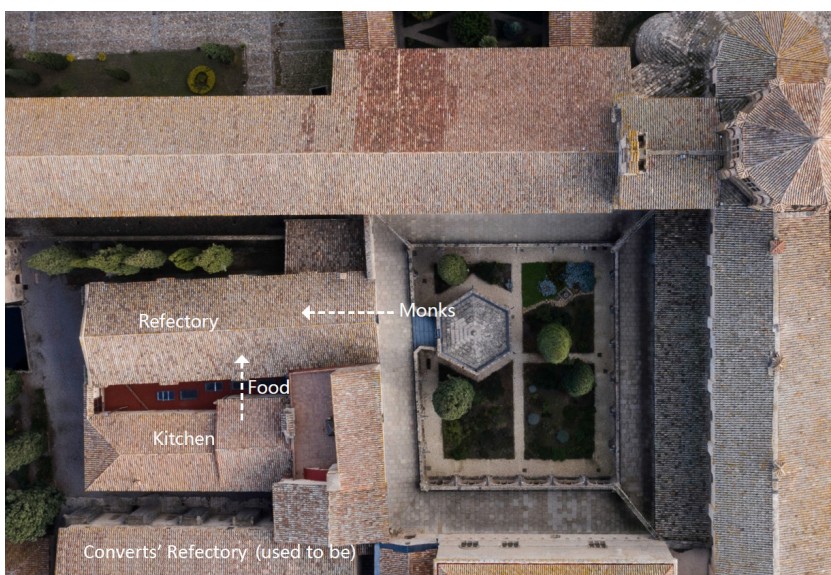

**Figure 34.** Aerial view of the Refectory of Poblet Monastery. Photo by the author.

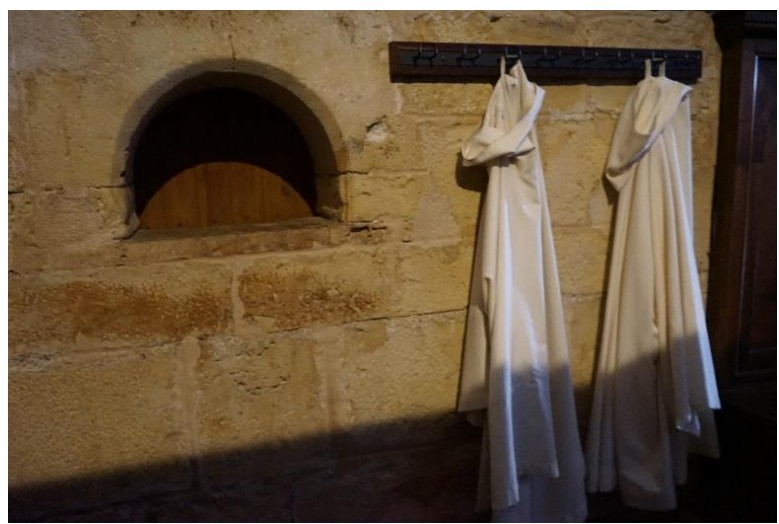

**Figure 35.** Hole for passing food between the Refectory and the old kitchen of Poblet Monastery. Photo by the author.

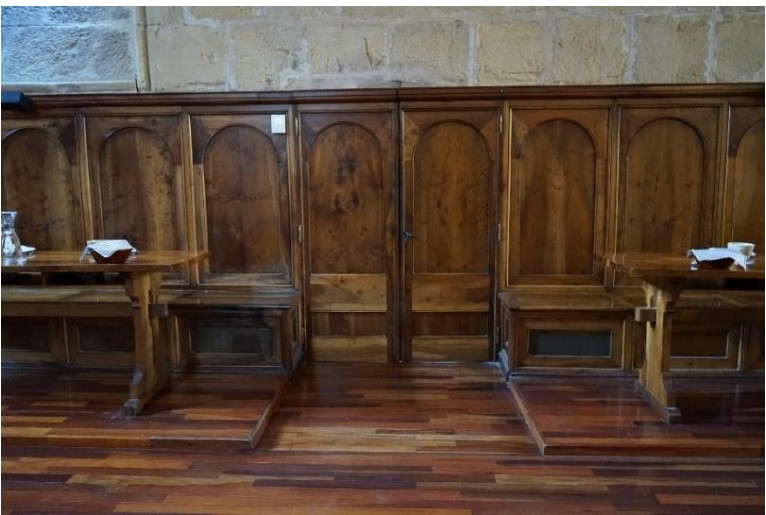

**Figure 36.** Wooden door toward the new kitchen of Poblet Monastery. Photo by the author.

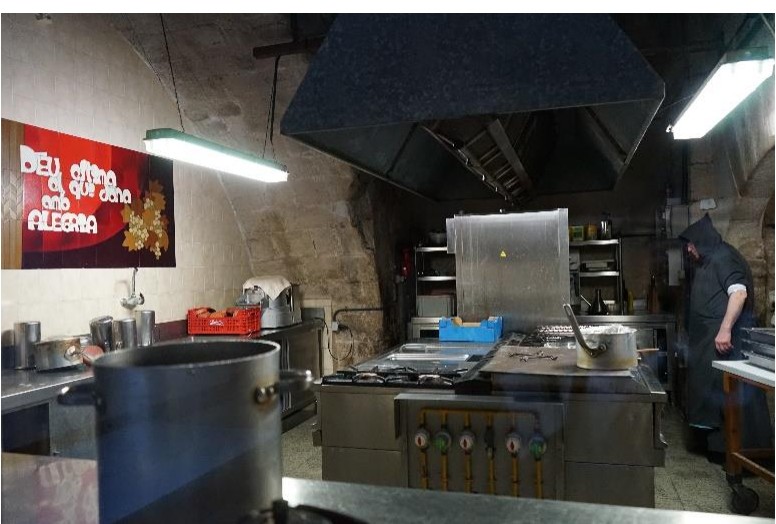

**Figure 37.** New kitchen of Poblet Monastery. Photo by the author.

In addition to valuing the efficiency of the dining process, efficient collaboration among different spaces of the logistics system is also fundamental in maintaining an orderly monastic life. In both monasteries, the logistics system space is closely connected to the logistics entrance of the monastery, making it convenient for monks to communicate with the outside world on a daily basis while also being far away from the space for worship so as to minimize any potential disruptions.

In the Poblet Monastery, the kitchen, Refectory, fountain pavilion (lavabo), and warming room are arranged together (Figure 38) to enhance energy efficiency, as well as to provide better service for the sacred Refectory. The conversi were the historical predecessors of the Cistercian Lay Brothers. On the one hand, the Lay Brothers sought to lead a life as close as possible to the monastic practices of the Cistercian Order, distancing themselves from secular life. On the other hand, they took on a significant amount of physical labor, assisting the Cistercian monks in their pursuit of self-sufficiency. Consequently, their relationship with the monks was both cooperative and distinct. By strategically placing the kitchen between the Refectory and the converts' Refectory, the architecture effectively planned their relationship, keeping them separated yet connected and maintaining a corresponding rhythm during mealtimes. In Guoqing Si, the Zhaitang (Monks' Dining Hall), Workers' Dining Hall, kitchen, and storage room (Granary) are arranged in a cluster near the logistics gate (Figure 39). This makes sure that the big old kitchen can serve the three dining halls at the same time.

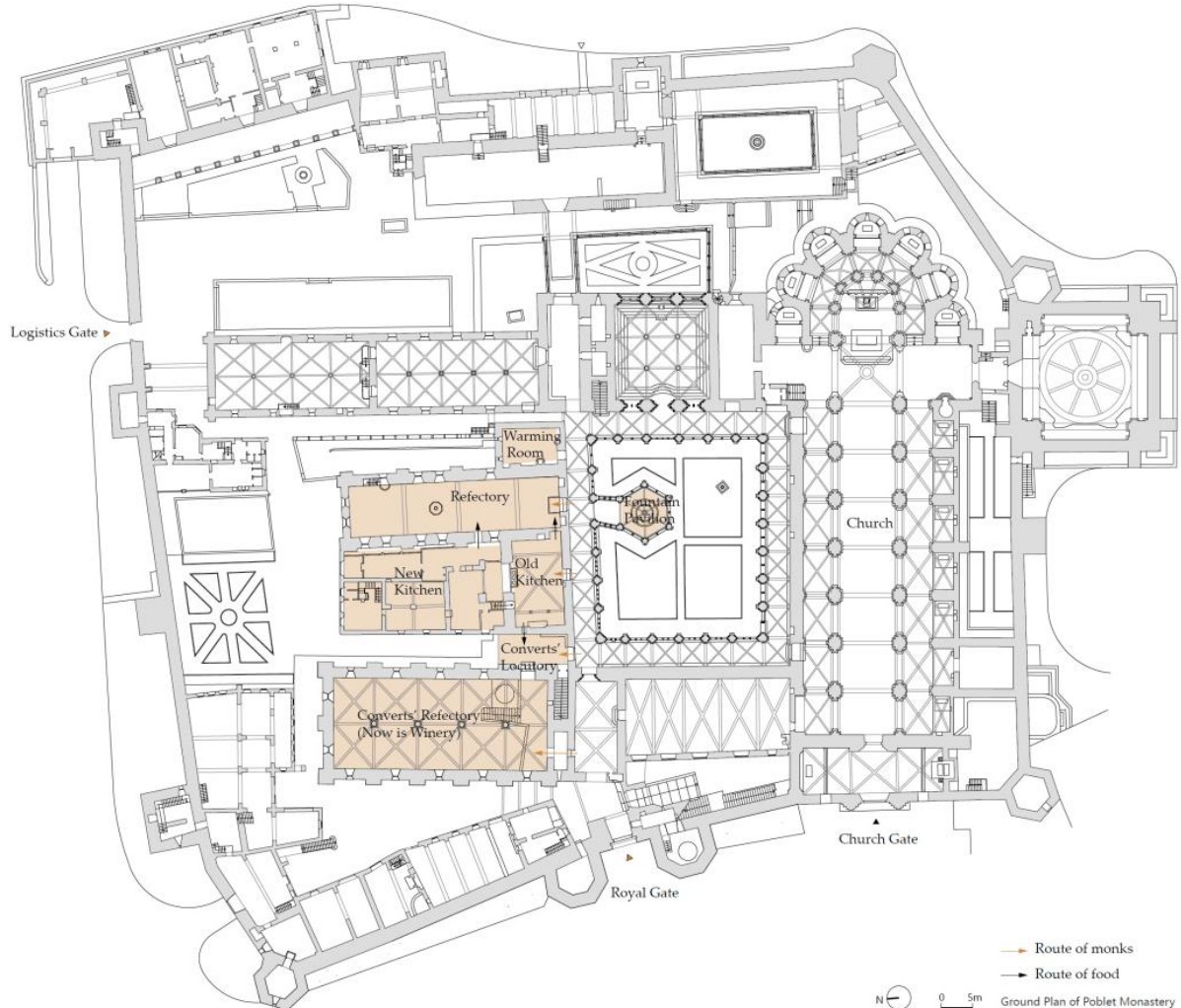

**Figure 38.** Logistics space system in Poblet Monastery. Drawn by the author.

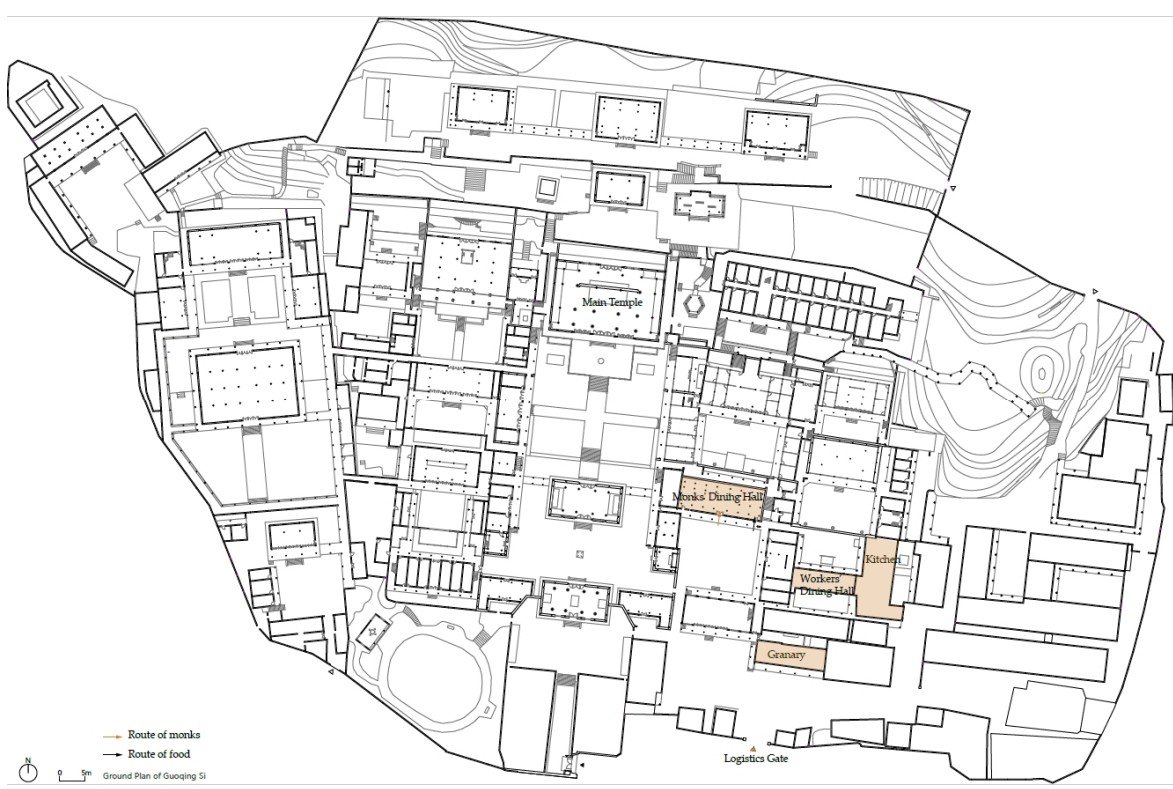

**Figure 39.** Logistics space system in Guoqing Si. Drawn by the author.

Regarding the location of the storage room in both monasteries, which serves as a prerequisite for ensuring safety, it observes the following arrangement principles: (1) Food is hidden in a location near a secondary entrance, away from the axis of worship (main entrances or the entrance of the church), but close to the residential area and the entrance/exit, making it closer to the fields where the harvest takes place. They are typically concealed behind solid walls with inconspicuous entrances. Even those residing in the living quarters can easily access the storage space. Paradoxically, the more public and prominent the location, the more secure it becomes. (2) Food (like rice) storage areas are placed in relatively dry or elevated locations, far from water and moisture, which is beneficial for preserving the food. (3) They are situated near the kitchen and dining hall, facilitating easy access for daily use. By following these guidelines, the storage of food in monasteries ensures both safety and convenience. It is worth mentioning that Guoqing Si mainly relies on rice as a staple food. Through the stairs to the second floor, new rice is poured into the granary, and then a small hole is opened at the bottom of the granary to retrieve the aged rice, which can effectively preserve the rice (Figure 40).

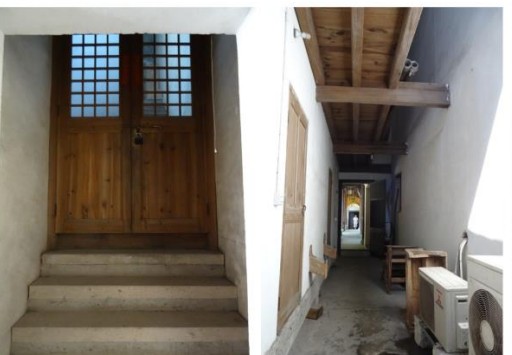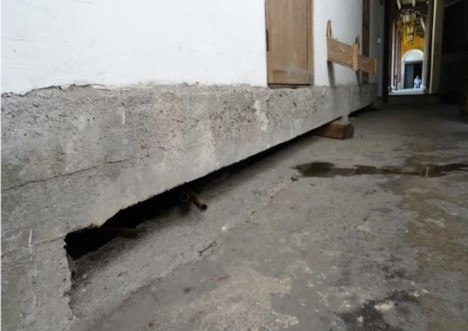

**Figure 40.** Rice storage room in Guoqing Si.

2.2.2. Cleanliness and Taboos

In both monasteries, the most sacred spaces are the main temple and the church, while the busiest daily spaces are undoubtedly the kitchen and dining areas. The latter spaces generate smoke, noise, food residue, and even odors in daily activities. Clearly, it is not dignified to hear sounds or smell food from the kitchen during prayers or religious ceremonies in the sacred halls. Therefore, it is necessary for the kitchen and dining areas to maintain a certain distance from the main temple or church to preserve the sanctity of the religious spaces.

Cistercian monasteries typically organize their spaces using a cloistered courtyard, with the Refectory situated opposite the church in a vertical arrangement. Although the Refectory's significance in the overall layout of the monastery is not as prominent as the main halls that it serves as a reference for the spatial arrangement (Wang 2021), the Refectory system is meticulous in its positioning within the logistical spaces and influences the specific layout of the area. The Refectory and kitchen are a fixed combination, but they are separate and independent structures. By maintaining this spatial arrangement, the monastery preserves the sanctity of the main halls while ensuring that the kitchen and dining areas can function efficiently without disturbing the religious practices conducted in the sacred spaces.

On the other hand, we can observe that vegetarian food, compared to meat, fish, and shellfish, tends to be cleaner in terms of food handling, less prone to producing unpleasant odors, and the eating process is simpler and easier to digest. From a deeper logical perspective, religious food taboos give a tangible spatial manifestation to the concept of sanctity. This manifestation is not only reflected in the interior design of dining spaces but also in their relationship with the overall architectural layout of the monastery. Religious food taboos reflect the belief in purity and holiness, and as a result, they influence the organization and arrangement of spaces within the sacred premises. The careful consideration given to the handling and consumption of food, particularly in relation to vegetarian options, highlights the significance of cleanliness and simplicity in the spiritual practice.

The importance of the purity and food layout is reflected in the following aspects: (1) Independence: having an independent dining space allows for focused food preparation and service, minimizing the risk of cross-contamination from other areas. It also provides a controlled environment where hygiene standards can be maintained effectively. (2) Complete Space: a well-designed logistics space system encompasses all the necessary facilities for food storage, preparation, cooking, and dining. This complete space ensures that every step of the food process can be performed hygienically and efficiently. (3) Order: the dining system promotes orderliness in the food service process. It includes organized serving areas, designated seating arrangements, and clear pathways for flow and movement. This not only enhances efficiency but also contributes to maintaining cleanliness by preventing overcrowding and confusion. By incorporating these principles into the food layout, purity is emphasized, leading to a silent and sacred daily practice through the act of eating.

*2.3. Food Structure and Monastic Landscape*

2.3.1. Self-Sufficiency

Self-sufficiency is an integral aspect of the monastic lifestyle, as monks seek to reduce their dependence on external resources and cultivate a sense of independence and simplicity. This means that they rely on their own labor to obtain ample supplies of water and food. By engaging in various activities such as farming, gardening, livestock rearing, and other forms of manual labor, monks aim to fulfill their basic needs directly from the resources available within their immediate surroundings. Obtaining sufficient sources of water (Wang and Feng 2023), whether from wells, springs, or other means, is crucial for their daily needs, such as drinking, cooking, and sanitation. Similarly, cultivating crops, tending to gardens, and raising animals provide them with a sustainable food source. Through self-sufficiency, monks develop a deeper connection with the natural world, foster a sense of gratitude for the provisions it offers, and embrace a humble and mindful existence. This

is not only a practical approach to sustaining their physical well-being but also an integral part of their spiritual journey.

In Guoqing Si, monks still maintain the tradition of self-sufficiency by growing their own grains and vegetables (Figure 41). However, they also rent out surplus land to farmers for cultivation. Additionally, they keep some cows to help plow the fields. It is worth mentioning that during harvest season, monks dry the rice in the courtyard (Figure 42). From this, we can see the importance of the courtyard in the agricultural era and its influence on the overall layout of the monastery. In the case of the Poblet Monastery, monks used to farm themselves (Figure 43). However, nowadays, they only keep a small part of the farms inside the monastery, with the rest outside the monastery being owned by farmers from surrounding villages. Before, the cultivated lands for monks included forests, vineyards, agricultural farms, fishing farms, and farms for livestock such as sheep, cattle, and horses. These farms may be located around the monastery or within a certain distance that can be reached in a day's walk.

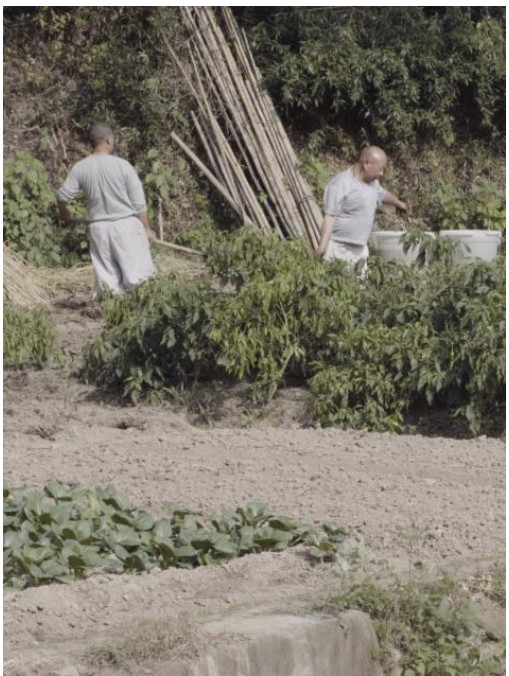

**Figure 41.** Monks of Guoqing Si work in the field. Photo by the author.

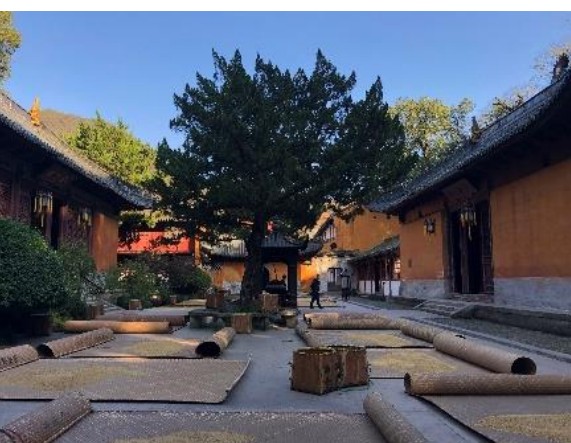

**Figure 42.** Monks dry the rice in the courtyard. Photo by the author.

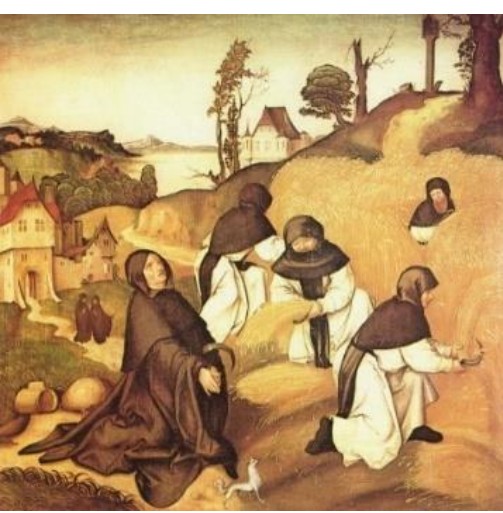

**Figure 43.** Cistercians at work in a detail from the Life of St. Bernard of Clairvaux. Illustrated by Jörg Breu the Elder in 1500, cited from https://commons.wikimedia.org/wiki/File:J%C3%B6rg_Breu_d._%C3%84._002.jpg. Accessed on 17 July 2023.

2.3.2. Wilderness Sanctuary

Both the Han Buddhist and the Cistercian lean toward a vegetarian meal. Although the Han Buddhists initially did not prohibit the consumption of clean meat, and the Cistercian still allows weak or sick monks to consume fish, overall, a vegetarian meal is more suitable for the monastery environment. It can contribute to a quiet atmosphere in meal preparation and dining spaces, as well as a serene and expansive surrounding environment.

In the case of Guoqing Si, located at the foot of the Bagui Peak (Figure 44), the monastery utilizes the open flat land on the southern side of the monastery to cultivate rice paddies (previously a release pond) (Figure 45) while developing vegetable gardens and fruit trees on the terraced fields on the western side of the monastery. This creates a picturesque scene of green rice paddies against the backdrop of the surrounding mountains when viewed from the foot of the mountain. At the Poblet Monastery, grapevines are predominantly cultivated around the monastery (Figure 46), with fields located within the monastery grounds (Figure 47). One can imagine that if the main food source for the monastery were poultry, pigs, or livestock, the environment would be filled with animal enclosures, the sounds of their calls, and the smell of their excrement, which would detract from the tranquility of the surroundings.

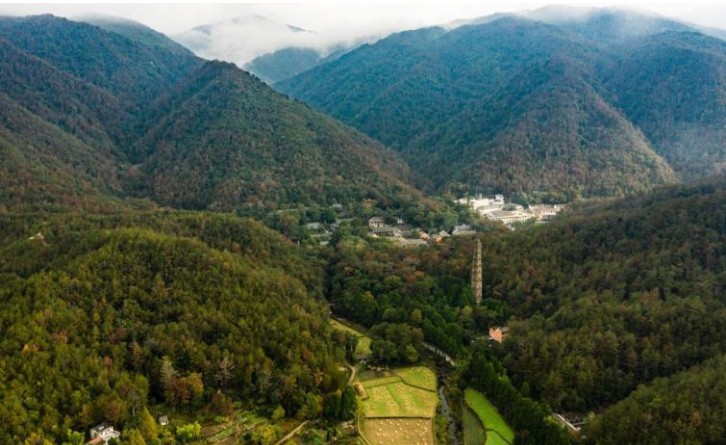

**Figure 44.** Surrounding environment of Guoqing Si. Photo by the author.

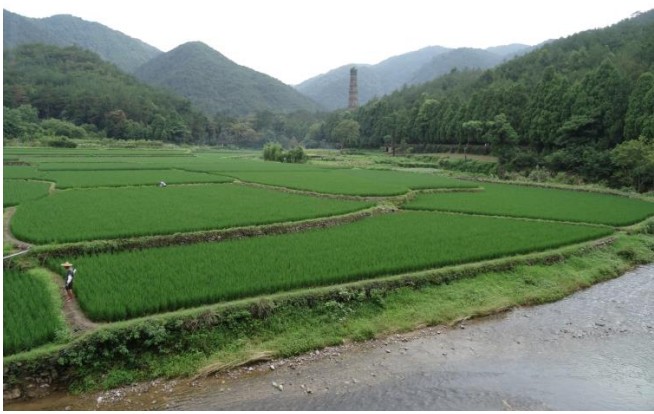

**Figure 45.** Rice paddies on the southern side of Guoqing Si. Photo by the author.

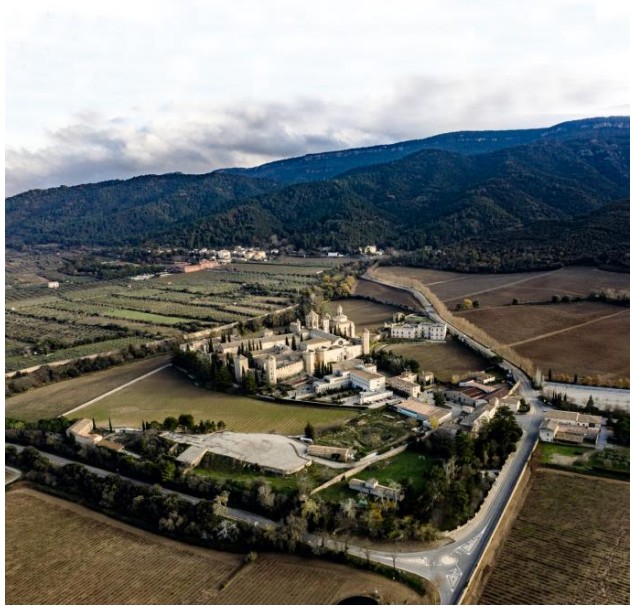

**Figure 46.** Environment surrounding Poblet Monastery. Photo by the author.

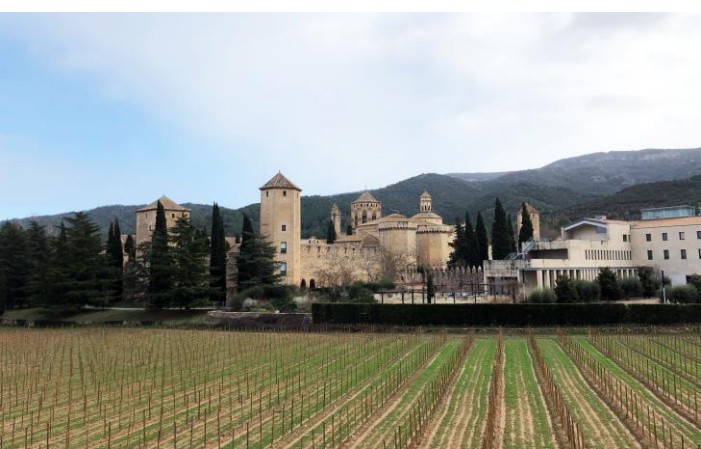

**Figure 47.** Grapevine fields inside Poblet Monastery. Photo by the author.

Therefore, it can even be said that the choice of food determines the nature of the sur-rounding environment. The emphasis on a vegetarian meal in these sacred spaces aligns with the principles of compassion, mindfulness, and harmony with nature, creating an environment that fosters peace and serenity.

*"It was within such untilled islets, which had emerged in the middle of irrigated fields, on uneven terrain, on mountains, in valleys, or on hillsides, that most of the monasteries were established … in most cases, the original kernel of the monastic lands was constituted from mountainous or hilly terrains".* (Gernet 1995, p. 117)

On the other hand, we also need to recognize that the picturesque locations where monasteries are situated today were not always so beautiful in the past. For monks, their land is typically acquired through donations. If they were to forcibly occupy land belonging to farmers, it would be difficult to obtain institutional protection. This means that they can only turn to uncultivated lands outside the existing land system. Therefore,

*"contrary to peasant properties that were all devoted to the cultivation of arable crops, the monastic estates-like those of the wealthy laity-were distinguished by the diversity of their farming: woods, copses, pastures, mountain gardens, and orchards there occupied a place of far greater importance than in the peasant economy".* (Gernet 1995, p. 117).

In Cistercian monasteries, things are similar. For the last period of reconquest, the Poblet Monastery was established for two purposes: religious expansion and regional repopulation. In that turbulent time, the establishment of the Cistercian monasteries had a very important role in stabilizing the lost land that had just been recovered from the Moorish. Poblet, situated in the frontier of reconquest, the west border of Barcelona, was expected to serve as an important point for attracting more population. Therefore, the Poblet Monastery, along with the Monastery of Santes Creus and Monastery of Vallbona de les Monges, are known as the Cistercian triangle (Figure 48), which helped consolidate power in Catalonia in the 12th century after the crown of Aragon was set.

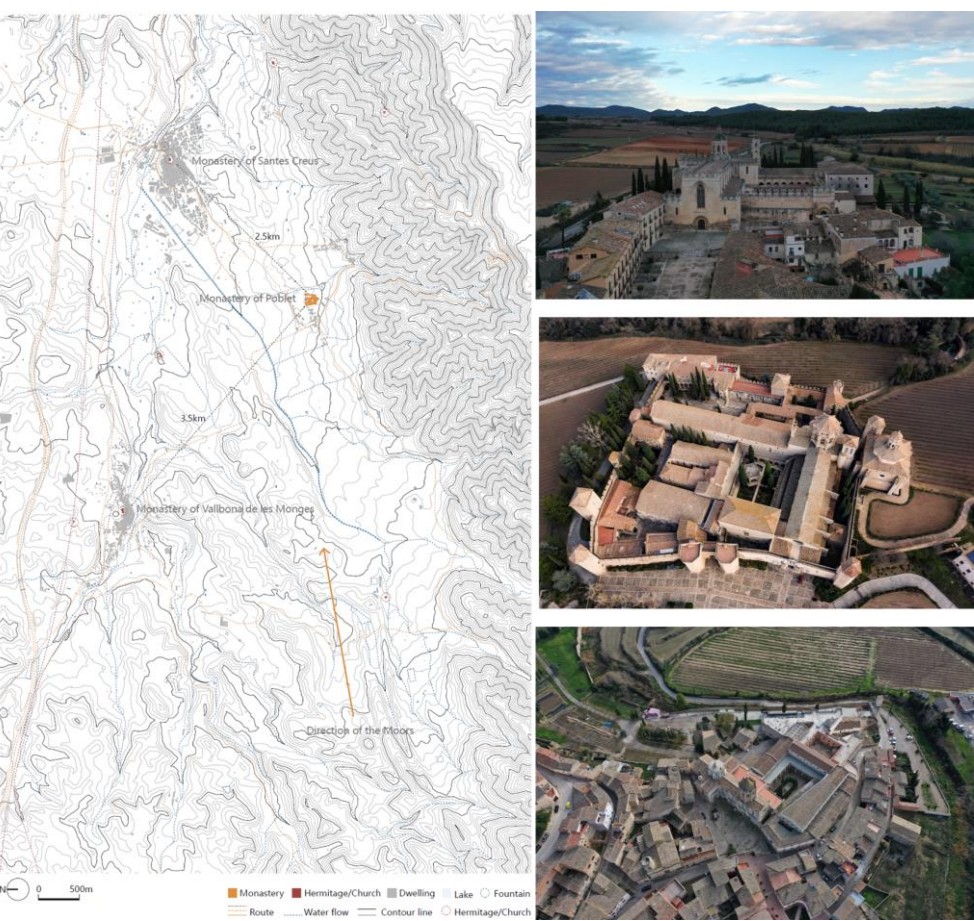

**Figure 48.** Cistercian Triangle in Catalonia. Drawn and photographed by the author.

The monks' pursuit of a self-sufficient monastic life and their choices of food based on monastic rules have ultimately influenced the formation of the surrounding landscape of the monastery. The plant- and tree-dominated environment has created a serene atmosphere for the monastery, ultimately transforming the once-barren wilderness into a sacred and picturesque sanctuary.

## 3. Discussion: Spatial Model of Food in Monasteries

After a detailed analysis of how food affects space formation, layout organization, and site selection, this study attempts to extract a spatial model of food in monasteries.

### 3.1. Sustainable Agri-Food Space System

The sustainable agri-food space system refers to a system that promotes sustainable agricultural practices and ensures the availability and accessibility of safe, nutritious, and sufficient food for all. It takes into account the environmental, economic, monastic lives, and spatial factors surrounding the monastery to ensure the long-term viability of the agricultural industry while minimizing negative impacts on the site's environment. It aims to minimize external dependencies and create a closed loop where food production, processing, and consumption are interconnected.

In the research by Goody (1982), he defined "the study of the process of providing and transforming food covers four main areas, that of growing allocating, cooking and eating, which represent the phases of production, distribution, preparation and consumption". Regarding cenobitic monastic life, a sustainable agri-food space system may include: (1) Food Production: both monasteries incorporate various methods of food production, such as cultivating vegetable gardens, fruit orchards, herb gardens, and aquaculture, as we can see from the ideal layout in Figures 49 and 50. (2) Waste Management: both of them emphasize efficient waste management, including composting organic waste and recycling food scraps to nourish the soil and enhance agricultural productivity. This closed-loop approach reduces waste and supports the sustainability of the food production cycle, as shown in the ideal layout where the toilet is located near the vegetable gardens. (3) Water Management: water conservation and management play a crucial role in the sustainability of agri-food space systems. Strategies such as rainwater harvesting, water recycling, and efficient irrigation techniques ensure that water resources are used efficiently and sustainably, as analyzed in another article (Wang and Feng 2023). (4) Food Processing: facilities for food processing, including warehouses for food storage, kitchens for food preparation, and dining areas for food consumption. The clean, efficient, and safe flow between them ensures the normal operation of the monastic space and distinguishes the sacred from the secular. (5) Meal-taking ritual: the sustainable agri-food space system promotes mindful and sustainable eating habits among monks in both monasteries.

By implementing a sustainable agri-food space system, monasteries can achieve greater self-sufficiency, reduce their ecological footprint, and foster a deeper connection with nature and the principles of sustainability. Indeed, upon seeing the ideal layouts of the food spatial system, some might say that it resembles the food processing methods in traditional villages, where food residue becomes fertilizer for the fields. It is true that in the Middle Ages, both Eastern and Western monks lived in agrarian societies. Monasteries functioned as small communal settlements, embodying the complete food-processing cycle and the industriousness that characterized human agricultural civilization.

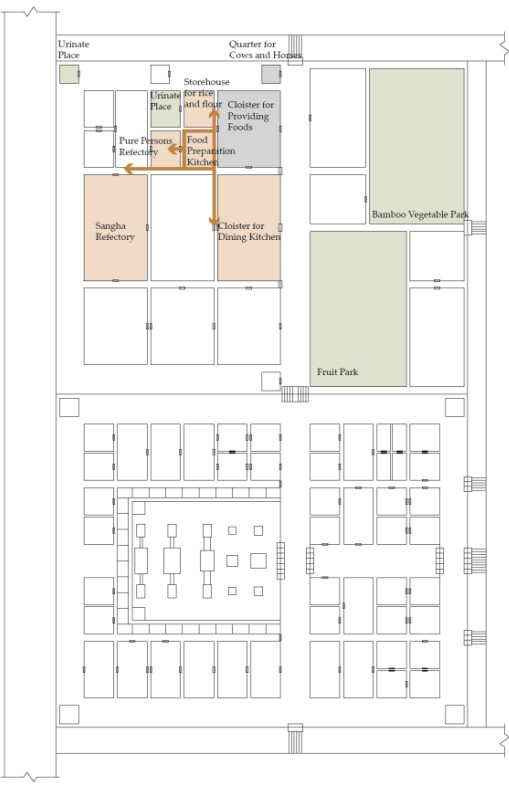

**Figure 49.** Agri-food space system for monks in the ideal layout of The Plan of Illustrated Scripture. Redrawn by the author.

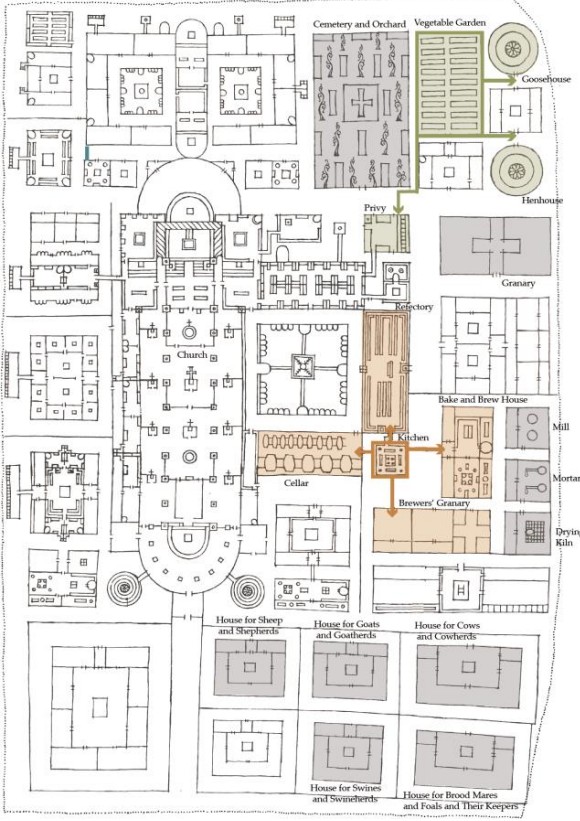

**Figure 50.** Agri-food space system for monks in the ideal layout of St. Gall Plan. Redrawn by the author.

### 3.2. A Distinct Food Spatial Order between the Sacred and the Secular Realms

*"If the proposed interpretation of the forbidden animals is correct, the dietary laws would have been like signs which at every turn inspired meditation on the oneness, purity and completeness of God. By rules of avoidance holiness was given a physical expression in every encounter with the animal kingdom and at every meal. Observance of the food rules would thus have been a meaningful part of the great liturgical act of recognition and worship which culminated in the sacrifice in the Temple".* (Douglas 1966, p. 58)

In Han Buddhist monasteries, food achieves its most sacred and symbolic meaning when it is placed in the center of the empty platform in front of the main hall (Figure 21). At this moment, it is not merely food, but it is intended for all sentient beings in the three realms of water, land, and air. Through the offering to the heavens ceremony, the original form of offering sacrifice expresses the respect of the sangha for the Buddha and the compassion for all sentient beings. The unity of food, the main hall, and the ceremony is the most thorough manifestation of the sangha's practice of Buddhism, and through this, they transcend the limitations of time and space. All six realms of sentient beings (heavenly, asura, human, animal, hungry ghost, and hell realms) are able to attend the vegetarian feast and obtain liberation, wisdom, and perfection.

*"The Christian Eucharist (Eucharist means "thanksgiving") was born directly from the Jewish Passover sacrifices … In this ritual, Christ, who is for believers both God and human, enters not only into the minds but into the bodies of the congregation; the people present at the table eat God. No animal and no new death is needed, no bridges required: God enters directly. The Eucharist is the ritual perpetuation of the incarnational relationship with humankind which God initiated through Christ (the word "incarnation" means "becoming flesh")".* (Visser 1991, p. 36)

In Cistercian monasteries, food, unleavened bread and wine, are the main components of the Catholic sacrament of Holy Communion (Figure 22). The establishment of the sacrament originated from Jesus breaking bread with his disciples during the Last Supper (a Jewish dietary tradition) and saying, "This is my body" and "This is my blood" when he passed around the wine. When pilgrims partake in unleavened bread and wine, they think of Jesus Christ being crucified for their sins and sacrificing himself for them. The sacrament ceremony, held in the church, is the climax of the food in the monastery, and its origin lies in its symbolism.

As discussed above, the spatial location and presentation of food in both monasteries are highly consistent with the distribution of the sacred and the secular in the monastic layout (Wang 2021). The combination of food and space presents the difference between the sacred and the secular through the role of the ritual. Visser (1991, p. 36) said,

*"The ceremony uses every psychological device defined by scholars of ritual. These include notions such as entrainment, formalization, synchronization, tuning and cognitive structuring, as well as spatial organization and focusing, and perfected ordinary action. Distances both temporal and spatial are collapsed, as ritual contact is made with past, present, and future at once, and as "this place" is united with "everywhere else", including the realm of the supernatural".*

Apart from the symbolic meaning of food, the spatial order of dining within a monastery is intricately linked to the distribution and structure of the sacred and secular spaces. The most sacred space, often the church or main hall, should not be disturbed by factors such as noise, smells, or water associated with food. Therefore, there is typically a distance and segregation maintained between the food system and sacred spaces. The dining area is not a standalone building but part of a comprehensive food processing system that includes storage, preparation, delivery, and consumption. This integrated system has logistical entrances, storage spaces, kitchens, serving corridors, and dining halls. The kitchen is where the basic daily functions of food provision begin, and then the food is consumed in a meditative manner in the dining hall. Finally, in the most important space, the main hall, the

food is offered as a form of worship. The spatial arrangement ensures that the reverence and sanctity of the monastery are preserved while also meeting the practical needs of food provision for the community, as Figures 51 and 52 show.

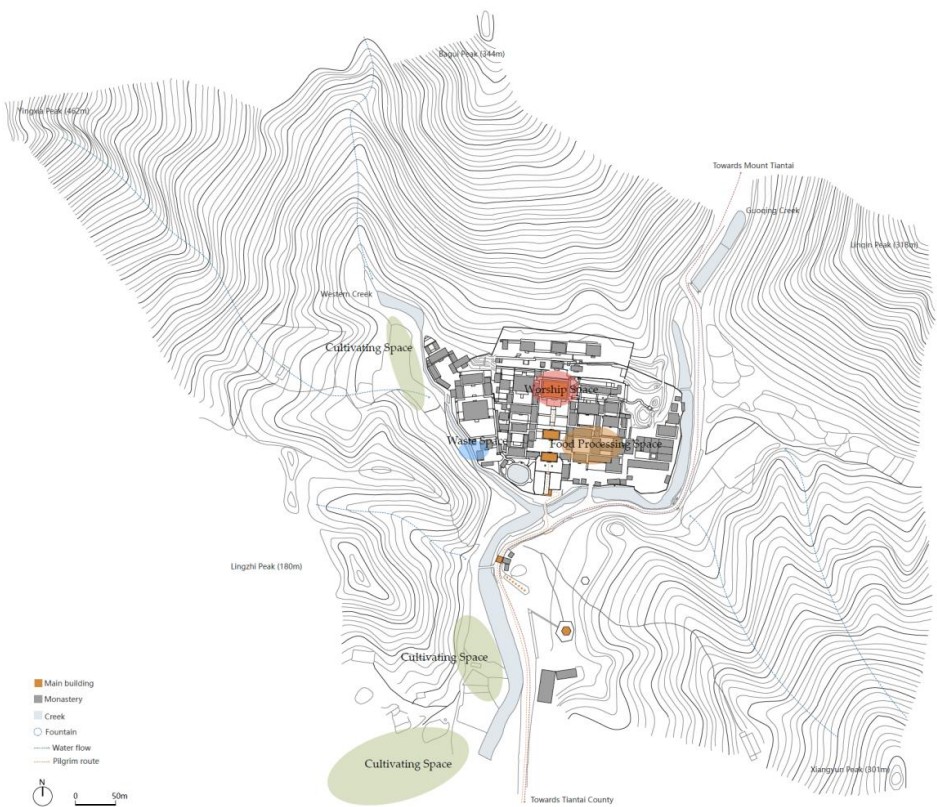

**Figure 51.** Food space layout of Guoqing Si. Drawn by the author.

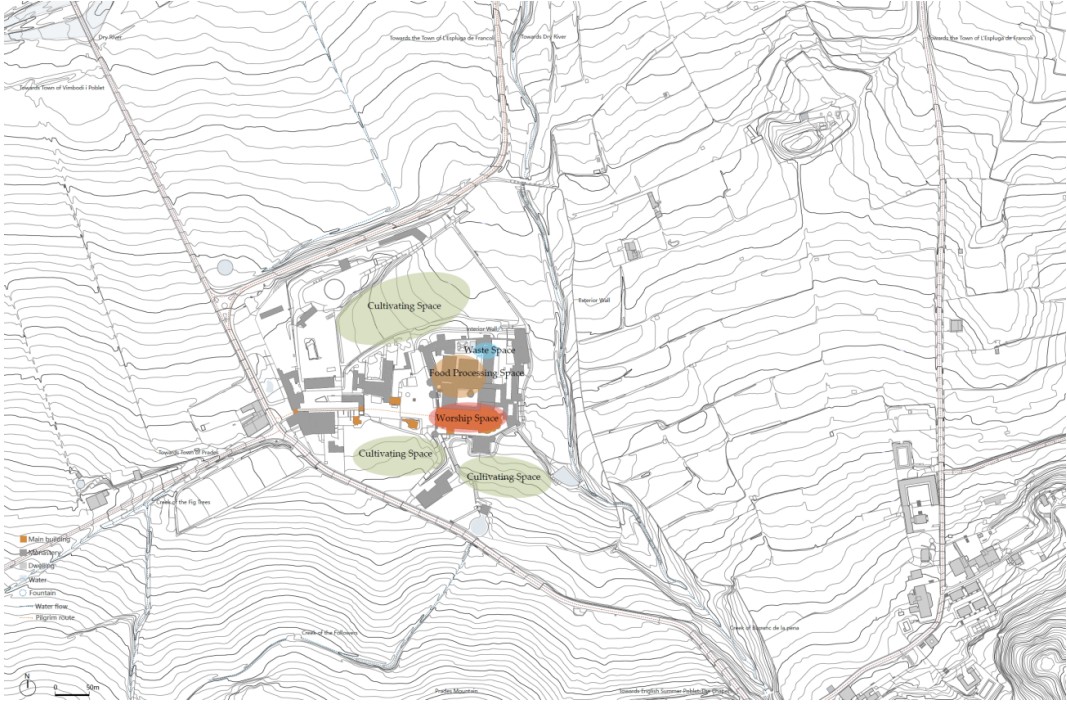

**Figure 52.** Food space layout of Poblet Monastery. Drawn by the author.

*3.3. Unusual Dining Space and Usual Meal*

In monasteries, the daily dining space may deviate from conventional norms, reflecting unique practices and food restrictions. These spaces are designed to accommodate the specific requirements and routines of monks, characterized by simplicity, mindfulness, and adherence to religious or spiritual principles.

The unusual aspects of daily dining spaces in such settings are mainly reflected in the following aspects: (1) Silent space: silence should be observed during meals to foster a sense of reflection and mindfulness. This practice allows individuals to focus on the act of eating and cultivate gratitude for the nourishment provided. Indeed, spacious and lofty spaces can amplify even the softest whispers and hushed conversations among individuals, including monks. The acoustics of large spaces can allow sound to travel further and linger, potentially disrupting the intended atmosphere of tranquility and contemplation. Both monasteries aim to create interior space that promotes silence and mindfulness to facilitate introspection, meditation, and spiritual practice. On the other hand, spacious environments can amplify the sound of prayers, especially when monks are reciting sacred texts or chanting during offerings. The reverberation of sound in such spaces can enhance the sense of sacredness during the dining process. The acoustics of large spaces can create a unique ambiance where the sound of prayers resonates and fills the air, creating a captivating and immersive experience for those present. The echoes and reverberations can add a spiritual dimension to the act of dining, deepening the connection between the individuals and their religious or contemplative practices. By amplifying the sound of prayers, these spacious environments can contribute to the overall spiritual ambiance, fostering a deeper engagement and connection with religious rituals and practices. (2) Ritualistic layout: the indoor layout and furniture arrangement of the Zhaitang and Refectory are prepared to create a sacred dining ritual. By utilizing the centrality of the architecture to arrange the position of the statues and the abbot and arranging the seats of the monks using the symmetrical spaces on both sides of the building, the importance of the central location is further emphasized, and the sacredness of the space is highlighted. In addition, by planning the food delivery route to come from the side door and the monks to enter from the main door, the sense of ritual in dining is further strengthened. The sacredness of the dining process is further elevated as individuals participate in communal recitation or chanting, creating a profound and meaningful experience for the community.

These spiritual and symbolic spaces are created to foster a deeper connection between individuals and their food, as well as their spiritual practices. They provide an environment where monks can engage in introspection, express gratitude, and cultivate a sense of mindfulness while partaking in nourishment. These spaces are designed to facilitate a heightened spiritual experience during the act of eating. The space envelops individuals and their food, while the overhead space is filled with sounds and light, further enhancing the sensory experience. The interplay of light, sound, and the surrounding environment creates an atmosphere conducive to spiritual reflection and inner growth. The concept of space in this context goes beyond physical dimensions and architecture. It encompasses the creation of a sacred atmosphere that transcends the mere consumption of food and invites individuals to connect with the divine, express gratitude, and deepen their spiritual journey. Through the thoughtful design and arrangement of these spiritual and symbolic spaces, the act of dining becomes a transformative experience, elevating the physical act to a profound spiritual encounter.

## 4. Conclusions

Food holds multifaceted meanings within different cultures and religions. It is not merely sustenance but also carries symbolic value, representing concepts such as nourishment, communion, sacrifice, and divine blessings. The act of dining within a monastic space becomes a profound expression of these symbolic meanings, connecting individuals with their religious beliefs, traditions, and practices. The symbolism and meaning of food play a crucial role in shaping the design and arrangement of spaces. The sound of

chanting, the use of light, and the overall ambiance all contribute to the creation of a sacred atmosphere that enhances the symbolic and spiritual dimensions of the dining experience. Additionally, cultural and religious practices influence the rituals and symbolism associated with dining in sacred spaces. These practices often prescribe specific behaviors, gestures, prayers, or blessings that further imbue the act of eating with profound meaning. The collective participation in these rituals fosters a sense of community, shared values, and a deep connection to the divine. The transformation of food from the mundane to the sacred within monastic spaces is a testament to the intricate interplay between culture, religion, and space. It underscores the significance of food as a vehicle for spiritual expression, cultural identity, and communal cohesion. Through this complex interrelationship, dining becomes a powerful medium through which monks engage with their religious beliefs, express devotion, and embody the core values and symbols of their faith.

The key difference between the Zhaitang or Refectory and a typical restaurant lies in the fact that their focus is not solely on making food the centerpiece of the dining experience. Instead, they aim to integrate the elements of food, body and mind, and ritual into a cohesive experience. Such high consistency of worship spaces can also be sensed in the caves with Buddhist motifs (Wang and Yan 2023). In the Zhaitang or Refectory, the act of dining goes beyond mere sustenance and becomes a holistic experience. It encompasses the spiritual, mental, and physical aspects of the individuals partaking in the meal. The emphasis is on creating a harmonious and balanced experience that nourishes not only the body but also the mind and spirit. This approach is often rooted in religious or monastic traditions, where the act of eating is viewed as an opportunity for reflection, gratitude, and connection with the divine. It is seen as a way to cultivate mindfulness, discipline, and a deeper understanding of oneself and one's relationship with the world.

> *"The power resting within outside meaning sets terms for the creation of inside, or symbolic, meaning … Objects, ideas, and persons take on a patterned structural unity in the creation of ritual"*. (Mintz 1996, pp. 30–31)

As Mintz (1996) described, the structural power of food in religious spaces transcends its everyday materiality and possesses symbolic significance, particularly within the context of religious and cultural meanings. To understand this outcome, we cannot overlook the restraining effect of religious precepts on monks, which, though constraints, transform into habits and become daily practices that reinforce the worship of God.

The impact of dining on space is reflected in space formation, layout organization, and site selection for spiritual practice, as well as the specific construction and arrangement of dining halls. These efforts aim to ensure the practicality of food, enhance the sacred experience of dining, and portray the religious imagery embedded in the act of eating. Under the guidance of a self-sufficient religious lifestyle, monks' food choices, influenced by the notions of purity and impurity in their religious beliefs, impact the types of food grown in the vicinity of the monastery. This, in turn, shapes the surrounding landscape, including orchards, rice fields, wheat fields, olive trees, grapevines, and water ponds. transforming it from wilderness into a sacred site and contributing to the overall impression of the spiritual venues.

> *"Granted that its (purity) root means separateness, the next idea that emerges is of the Holy as Wholeness and completeness"*. (Douglas 1966, p. 52)

Ultimately, monks are constantly engaged in their spiritual practices, and the highest level is achieved when the act of eating, the ritual of worship, the discipline of spiritual practice, and the dining space seamlessly merge into one. The daily and spiritual aspects become inseparable, and through the influence of food, the transition from the mundane to the sacred is realized within the space.

**Funding:** This research and the APC was funded by Shanghai Pujiang Programme (23PJC102).

**Institutional Review Board Statement:** Not applicable.

**Data Availability Statement:** No new data were created or analyzed in this study. Data sharing is not applicable to this article.

**Acknowledgments:** This article is sponsored by Shanghai Pujiang Programme (23PJC102), and it is conducted under the financial support from the China Scholarship Council, 2014–2018, File No. 201406260218. It derives from the dissertation under the direction of José Ignacio Linazasoro and Fangji Wang. I am very grateful to them for their detailed guidance on the research of the article. Besides, thanks to Yan Liu who have provided constructive comments on the article.

**Conflicts of Interest:** The author declares no conflict of interest.

## Note

1   "七食須觀門五別。一計功多少。量彼來處。二自忖己德行全缺多減。三防心顯過不過三毒。四正事良藥取濟形若。五為成道業世報非意" (T40n1804 1988, p. 115)。但后人多沿用宋·黃庭堅所著[士大夫食时五观]:"一、計功多少，量彼來處。二、忖己德行，全缺應供。三、防心離過，貪等為宗。四、正是良藥，為療形苦。五、為成道業，故受此食"。宋山谷黃庭堅著，明梅墟周履靖校《讀北山酒經客談》卷二，〈士大夫食時五觀〉。

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
