# Peer review of "Food and Monastic Space: From Routine Dining to Sacred Worship—Comparative Review of Han Buddhist and Cistercian Monasteries Using Guoqing Si and Poblet Monastery as Detailed Case Studies"

_religions, doi:10.3390/rel15020217_

Round 1

Reviewer 1 Report

Comments and Suggestions for Authors

Foot of figure 39 refers to Guonqing Si, not to Poblet

Comments on the Quality of English Language

To my knowledge English is fine. Only minor spelling mistakes to be checked. I've seen:

-l. 198 a verb is missing, probably "reflect"

-l. 715 Santes Creus (not Crues) and Vallbona de les Monges (not Monge) (these are Catalan words)

-fig. 19 it's the german Diet, not "Meal" (maybe from authomatic translation)

Author Response

Dear Reviewer,

Thanks for your comments and kind suggestions.

-Foot of figure 39 refers to Guonqing Si, not to Poblet

Thank you so much for pointing out the mistake, I have corrected the foot of figure 39 in the manuscript.

-

Comments on the Quality of English Language

To my knowledge English is fine. Only minor spelling mistakes to be checked. I've seen:

-l. 198 a verb is missing, probably "reflect"

Thanks for your kind observation. Here I attach the original complete sentences and I also add it to the reference, hoping that from the complete context, misunderstandings can be avoided.

“Thus fountains became receptacles of living water, dorters became chambers of sleep, chapter-houses enshrined the gravity and solemnity of chapter-sittings, and refectories the importance ascribed to the common meal in the regimen of ascetics. The meagre fare was eaten in princely dining-halls, which sometimes rivalled churches in their size and magnificence.”

-l. 715 Santes Creus (not Crues) and Vallbona de les Monges (not Monge) (these are Catalan words)

Thank you so much for pointing out the mistake, I have revised the error of the word.

-fig. 19 it's the german Diet, not "Meal" (maybe from authomatic translation)

Thank you so much for pointing out the mistake, I have revised the error of the word.

Reviewer 2 Report

Comments and Suggestions for Authors

This is an excellent article. It is well presented, very well structured and explained, and innovative in its material, analysis and conclusions. The article sets out to demonstrate a connection between the theoretical (spiritual) ideals of two types of monastic communities with food production, consumption and dining rituals. The author chooses two very different case studies (Guoqing Si Hahn Buddhist Monastery and Poblet Cistercian Monastery) and shows how the model for analysis runs equally for both. The author treats both the architectural space of the dining areas and the surrounding supporting buildings, looks at such issues as self sufficiency and independence of the dining building and at the ideologies behind production and consumption. The article looks at wide circles such as the environment around the monasteries and is supported by very rich photo documentation from both monasteries and by well-made and clear architectural drawings produced by the author. The article demonstrates well the ways in which the design of the space and the ceremonial aspects of the meals align with ideas such as “compassion, mindfulness, and harmony with nature, creating 692 an environment that fosters peace and serenity” (p. 30). The sociological/anthropological, theological and phenomenological aspects are well explained. Less prominent in the paper is the historical prism. The date and circumstances of foundation of the two edifices arrive late in the paper. This is also reflected in the bibliography which could be richer, especially in history of architecture, Cistercian architectural ideals, the ideal plan of St. Gall found in the paper etc. Therefore, while the case studies are well chosen and very closely documented and analysed, the supporting literature is too slim in my opinion. The author sets out to bridge the fields of anthropology, sociology and architecture but these are historic buildings so there is another gap to bridge - between the context of original design and construction and the realities of the monks today. Historical supporting evidence therefore (such as the rule of St. Benedict) should appear with a date and a word about how central the text continues to be for the monks today. I think the excellent and innovative article should be accepted but would hope for more use of specific bibliography on religious architecture (rather than comparative religion).

Smaller points: 

p. 5 - I think a word is missing in the quote (“Refectories the importance”).

p. 9 - I would imagine the fountain pavilion is later than the foundation date of the monastery. If so this can be stated. There should be reference to at least one foundational study on each building (other than any publications by the authors themselves). 

p. 19 Figure 29 - the image says ‘Zhaitang’ but I believe this is the refectory. 

p. 20 Figure 30 - The source itself (the plan) should be dated, there should be a reference to the archive or library and not an internet site. The historical circumstances of the plan are relevant here. 

Author Response

Dear Reviewer,

Thank you so much for your encouragement and valuable suggestions. I wholeheartedly agree with your points. For these two historical buildings and important cultural heritage sites, their historical significance cannot be overlooked. On the one hand, I have already analyzed their architectural history in previously published articles, so this article will not repeat that discussion. On the other hand, for this article, the correspondence between life and space, influenced by religious regulations, possesses a timeless quality, serving as a core principle that withstands the test of history. It does not undergo significant changes over time, and this correspondence is further divided into three aspects: space formation, layout organization, and site selection, which are the key focus of this study. Therefore, this article does not dwell extensively on architectural history analysis. However, your points are very valid, and I will provide additional explanations in the article. Below are some paragraphs with amendments. The specific modifications will appear in the manuscript in track changes mode.

Page3

These reasons support the selection of Han Buddhist and Cistercian monasteries as research objects and provide a foundation for understanding the impact of food on the space formation, layout organization, and site selection.

Furthermore, Guoqing Si and Poblet Monastery will serve as exemplary cases for studying the relationship between food and space. It should be noted that as important architectural cultural heritage, they have evolved continuously over a long historical period, evident in their architectural scale and specific spatial arrangements. Given that previous scholars have provided detailed records of the history and architectural evolution of both monasteries (Finestres y de Monsalvo, 1947) (Altisent,1974) (Guanding, n.d.) (Ding, 1995), and the author has conducted in-depth analyses of their spatial layout and the ideal plans referenced (Wang, 2021) (Wang, 2023), this article will not repeat the discussion. On the other hand, for this study, the correspondence between life and space is the main research focus. Due to the stable influence of reli-gious regulations, this correspondence possesses permanence, serving as a core princi-ple that withstands the test of history. Therefore, analyses and theoretical models can be established from the perspectives of site selection, layout, and spatial configuration. This spatial theoretical model will not undergo significant changes over time, espe-cially when viewed from the perspective of human dietary habits. Despite significant technological and societal changes, human needs for food maintain a relatively simple relationship with nature.

Therefore, the study aims to establish a spatial model of food in monasteries, de-picting the complete relationship between the cognition of food and the formation of monastic space, and analyzing how dining is transformed from daily routine to sacred worship.

Page 4

Very similar to Han Buddhists, according to the Rule of St. Benedict, set by an Italian abbot, Benedict of Nursia in the sixth century while still fundamental in monks’ daily life, Cistercian monks also adhere to a simple lifestyle to prevent their hearts from becoming “weighted down with surfeiting” (Rule of St. Benedict).

Page 19

In the ideal plan (Figure 30), the earliest surviving plan of a complete Benedictine monastery drawn in 816, the refectory is parallel to the church, and above it is the vestiary (cloister), closely connected to the dormitory (Wang, 2023).

Smaller points: 

P.5 - I think a word is missing in the quote (“Refectories the importance”).

Thanks for your kind observation. Here I attach the original complete sentences and I also add it to the reference, hoping that from the complete context, misunderstandings can be avoided.

“Thus fountains became receptacles of living water, dorters became chambers of sleep, chapter-houses enshrined the gravity and solemnity of chapter-sittings, and refectories the importance ascribed to the common meal in the regimen of ascetics. The meagre fare was eaten in princely dining-halls, which sometimes rivalled churches in their size and magnificence.”

P.9 - I would imagine the fountain pavilion is later than the foundation date of the monastery. If so this can be stated. There should be reference to at least one foundational study on each building (other than any publications by the authors themselves). 

Just in front the Refectory, there is a fountain pavilion constructed at the same time with the Refectory (Figure 15) where monks used to wash their hands before entering the Refectory after working in the fields. 

P.19 Figure 29 - the image says ‘Zhaitang’ but I believe this is the refectory. 

Thank you so much for pointing out the mistake, I have revised the error of the word.

P.20 Figure 30 - The source itself (the plan) should be dated, there should be a reference to the archive or library and not an internet site. The historical circumstances of the plan are relevant here. 

Thanks for your kind suggestion and very important opinion, I have added historical reference for the plan.